# Antisense, but not sense, repeat expanded RNAs activate PKR/eIF2α-dependent ISR in *C9ORF72* FTD/ALS

Janani Parameswaran[1], Nancy Zhang[1], Elke Braems[2,3], Kedamawit Tilahun[1], Devesh C Pant[1], Keena Yin[1], Seneshaw Asress[4], Kara Heeren[2,3], Anwesha Banerjee[1], Emma Davis[1], Samantha L Schwartz[5], Graeme L Conn[5], Gary J Bassell[1], Ludo Van Den Bosch[2,3], Jie Jiang[1]*

[1]Department of Cell Biology, Emory University, Atlanta, United States; [2]Department of Neurosciences, Experimental Neurology and Leuven Brain Institute, KU Leuven, Leuven, Belgium; [3]Center for Brain & Disease Research, Laboratory of Neurobiology, VIB, Campus Gasthuisberg, Leuven, Belgium; [4]Department of Neurology, Emory University, Atlanta, United States; [5]Department of Biochemistry, Emory University, Atlanta, United States

*For correspondence:
jie.jiang@emory.edu

Competing interest: The authors declare that no competing interests exist.

**Abstract** GGGGCC ($G_4C_2$) hexanucleotide repeat expansion in the *C9ORF72* gene is the most common genetic cause of frontotemporal dementia (FTD) and amyotrophic lateral sclerosis (ALS). The repeat is bidirectionally transcribed and confers gain of toxicity. However, the underlying toxic species is debated, and it is not clear whether antisense CCCCGG ($C_4G_2$) repeat expanded RNAs contribute to disease pathogenesis. Our study shows that *C9ORF72* antisense $C_4G_2$ repeat expanded RNAs trigger the activation of the PKR/eIF2α-dependent integrated stress response independent of dipeptide repeat proteins that are produced through repeat-associated non-AUG-initiated translation, leading to global translation inhibition and stress granule formation. Reducing PKR levels with either siRNA or morpholinos mitigates integrated stress response and toxicity caused by the antisense $C_4G_2$ RNAs in cell lines, primary neurons, and zebrafish. Increased phosphorylation of PKR/eIF2α is also observed in the frontal cortex of *C9ORF72* FTD/ALS patients. Finally, only antisense $C_4G_2$, but not sense $G_4C_2$, repeat expanded RNAs robustly activate the PKR/eIF2α pathway and induce aberrant stress granule formation. These results provide a mechanism by which antisense $C_4G_2$ repeat expanded RNAs elicit neuronal toxicity in FTD/ALS caused by *C9ORF72* repeat expansions.

## Editor's evaluation

The current study provides important, mechanistic insight into the potential contribution of antisense C4G2 expanded RNA to disease in C9orf72-associated ALS/FTD. The authors convincingly demonstrate that expression of this RNA species activates the PKR/eIF2α-dependent integrated stress response. They further provide evidence that this can contribute to disease phenotypes using multiple models and post-mortem patient samples.

## Introduction

In 2011, GGGGCC ($G_4C_2$) hexanucleotide repeat expansion in the first intron of the chromosome 9 open-reading frame 72 (*C9ORF72*) gene was identified as the most common genetic cause of frontotemporal dementia (FTD) and amyotrophic lateral sclerosis (ALS), two neurodegenerative diseases

that are now believed to belong to a continuous disease spectrum with clinical, pathological, and genetic overlaps (*DeJesus-Hernandez et al., 2011*; *Renton et al., 2011*). In normal populations, the $G_4C_2$ repeat size is between 2 and 30, whereas it expands to hundreds or thousands in FTD/ALS patients (referred to hereafter as C9FTD/ALS). C9FTD/ALS thus joins an increasing number of repeat expansion disorders, including Huntington's disease, myotonic dystrophy, and several spinocerebellar ataxias (*La Spada and Taylor, 2010*). Based on the initial pathological assessment of C9FTD/ALS patient postmortem tissues and lessons learned from other repeat expansion disorders, several pathogenic mechanisms by which the expanded *C9ORF72* repeats can exert toxicity were proposed (*Jiang and Ravits, 2019*). First, expanded $G_4C_2$ repeats inhibit *C9ORF72* mRNA transcription, leading to haploinsufficiency of the C9ORF72 protein (*Braems et al., 2020*; *Gijselinck et al., 2012*; *van Blitterswijk et al., 2015*); second, *C9ORF72* repeats are bidirectionally transcribed into sense $G_4C_2$ and antisense CCCCGG ($C_4G_2$) RNAs. These repeat expanded RNAs may cause gain of toxicity by sequestering essential RNA-binding proteins (RBPs) into RNA foci and/or by the production of toxic dipeptide repeat (DPR) proteins via non-canonical repeat-associated non-AUG-dependent (RAN) translation from all reading frames. More specifically, translating from sense $G_4C_2$ RNAs produces GA (glycine-alanine), GP (glycine-proline), and GR (glycine-arginine) DPR proteins, and translating from antisense $C_4G_2$ RNAs produces GP (glycine-proline), PA (proline-alanine), and PR (proline-arginine) DPR proteins (*Ash et al., 2013*). In addition to these pure dimeric DPR proteins, there is also evidence of chimeric DPR proteins in both model systems and patients (*Gao et al., 2017*; *McEachin et al., 2020a*; *Tabet et al., 2018*).

How *C9ORF72* repeat expansions cause FTD/ALS has been extensively explored. Although reducing C9ORF72 in zebrafish or *Caenorhabditis elegans* can cause motor deficits (*Swinnen et al., 2018*; *Therrien et al., 2013*), reduced or even complete deletion of C9ORF72 in mice does not lead to FTD/ALS-like abnormalities, suggesting that loss of C9ORF72 is not the main disease driver (*Atanasio et al., 2016*; *Burberry et al., 2016*; *Koppers et al., 2015*; *O'Rourke et al., 2016*; *Sudria-Lopez et al., 2016*; *Sullivan et al., 2016*; *Ugolino et al., 2016*). Supporting this, no missense or truncation mutations in *C9ORF72* are yet found in FTD/ALS patients (*Harms et al., 2013*). On the other hand, several lines of studies, by expressing either $G_4C_2$ repeats (*Chew et al., 2015*; *Jiang et al., 2016*) or individual codon-optimized, ATG-driven DPR proteins (*Choi et al., 2019*; *Mizielinska et al., 2014*; *Zhang et al., 2019*), support that gain of toxicity from the repeat expanded RNAs plays a central role in disease pathogenesis. Finally, loss of C9ORF72, which plays a role in autophagy/lysosomal functions, can exacerbate toxicity from the repeat expanded RNAs (*Boivin et al., 2020*; *Zhu et al., 2020*).

The underlying toxic species arising from *C9ORF72* repeat expanded RNAs that drive disease is still debated. Several RBPs have been suggested to interact with $G_4C_2$ or $C_4G_2$ repeat RNAs and to co-localize with RNA foci (*Celona et al., 2017*; *Conlon et al., 2016*; *Donnelly et al., 2013*; *Haeusler et al., 2014*; *Lee et al., 2013*; *Mori et al., 2013*; *Sareen et al., 2013*; *Xu et al., 2013*). However, strong evidence supporting that loss of any proposed RBPs drives C9FTD/ALS is lacking. In contrast, ectopic expression of individual DPR proteins, especially GR and PR, causes toxicity in various model systems (*Boeynaems et al., 2016*; *Choi et al., 2019*; *Freibaum et al., 2015*; *Hao et al., 2019*; *Kanekura et al., 2016*; *Lee et al., 2016*; *May et al., 2014*; *Mizielinska et al., 2014*; *Schludi et al., 2017*; *Tao et al., 2015*; *Wen et al., 2014*; *Yamakawa et al., 2015*; *Yang et al., 2015*; *Zhang et al., 2018*; *Zhang et al., 2016*; *Zhang et al., 2019*; *Zhang et al., 2014*; *Zu et al., 2011*). To determine the relative contributions of RNA foci- and DPR protein-mediated toxicity, two studies employed interrupted repeats with stop codons in all reading frames to prevent DPR protein production and concluded that neither sense or antisense repeat expanded RNAs are toxic in *Drosophila* (*Moens et al., 2018*; *Tran et al., 2015*). This was challenged by another study showing both sense and antisense repeat expanded RNAs can cause motor axonopathy in zebrafish independent of DPR proteins (*Swinnen et al., 2018*). Irrespective of RNA foci and DPR proteins, studies using antisense oligonucleotide (ASOs) to selectively degrade sense $G_4C_2$ repeat expanded RNAs strongly support its role in C9FTD/ALS pathogenesis. These sense strand-specific ASOs not only mitigate toxicity from *C9ORF72* repeat expansions in both transgenic mice expressing $G_4C_2$ repeats (*Jiang et al., 2016*) and patient IPSC-derived neurons (*Donnelly et al., 2013*), but also reverse downstream cellular and molecular alterations such as nucleocytoplasmic transport deficits (*Zhang et al., 2015*). However, whether antisense $C_4G_2$ repeat expanded RNAs contribute to C9FTD/ALS and thus are targets of intervention is less clear. Although PR translated from the antisense strand is extremely toxic in model systems, PR

or its aggregates are rare in postmortem tissues. Antisense RNA transcripts are also hard to detect. Surprisingly, several studies showed that antisense RNA foci are as abundant as sense RNA foci in multiple brain regions (*DeJesus-Hernandez et al., 2017*; *Mizielinska et al., 2013*; *Vatsavayai et al., 2019*), raising a possibility that antisense $C_4G_2$ repeat expanded RNAs also contribute to C9FTD/ALS (*McEachin et al., 2020b*). In this study, we show that antisense *C9ORF72* $C_4G_2$ repeat expanded RNAs are neurotoxic independent of RAN translated DPR proteins. Antisense $C_4G_2$, but not sense $G_4C_2$, repeat expanded RNAs activate PKR/eIF2α-dependent integrated stress response, leading to global protein synthesis inhibition and stress granules formation. Moreover, the phosphorylation of PKR/eIF2α is significantly increased in C9FTD/ALS patients, suggesting that antisense $C_4G_2$ repeat expanded RNAs contribute to disease pathogenesis.

## Results

### *C9ORF72* antisense $C_4G_2$ expanded repeats are neurotoxic

To determine the contribution of *C9ORF72* antisense repeat expanded RNAs in FTD/ALS pathogenesis, we first generated a construct containing 75 $C_4G_2$ repeats using recursive directional ligation as previously described (*Mizielinska et al., 2014*). We included six stop codons (two every frame) at the N-terminus to prevent unwarranted translation initiation and three protein tags in frame with individual DPR proteins at the C-terminus (*Figure 1A*). Recent studies have shown that the nucleotide sequences at 5′- and 3′- regions of expanded repeats regulate toxicity (*He et al., 2020*; *Sellier et al., 2017*). Although the molecular mechanism of *C9ORF72* antisense transcription initiation is unknown, it has been shown that transcription can start from at least 450 bp nucleotides upstream (*Zu et al., 2013*). We therefore added 450 bp of human sequence at the 5′- region of the antisense $C_4G_2$ repeats and termed this construct as 'in_($C_4G_2$)75'. When expressed in HEK293T cells, we detected an abundant accumulation of antisense RNA foci, but not in control cells expressing 2 $C_4G_2$ repeats (*Figure 1B*). Using antibodies against individual DPR proteins RAN translated from $C_4G_2$ expanded repeat RNAs or the protein tags in frame, we also observed production of GP, PR, and PA DPR proteins only in cells expressing in_($C_4G_2$)75 but not 2 repeats (*Figure 1—figure supplement 1A and B*). Antisense RNA foci and DPR proteins were also observed in mouse primary cortical neurons expressing in_($C_4G_2$)75, but not in neurons expressing 2 repeats (*Figure 1B* and *Figure 1—figure supplement 1C*). Thus, in_($C_4G_2$)75 produces antisense RNA foci and DPR proteins, two cellular pathological hallmarks observed in C9FTD/ALS patients.

To determine whether *C9ORF72* antisense $C_4G_2$ expanded repeats can cause neuronal toxicity, we co-transfected in_($C_4G_2$)75 or control 2 repeats together with mApple in mouse primary cortical neurons at 4 days in vitro (DIV4) and used automated longitudinal microscopy to track over days the survival of hundreds of neurons as indicated by the mApple fluorescence. Neurons expressing in_($C_4G_2$)75 die much faster than those expressing control 2 repeats, suggesting that *C9ORF72* antisense $C_4G_2$ expanded repeats are neurotoxic (*Figure 1C*).

### *C9ORF72* antisense $C_4G_2$ expanded repeats activate PKR/eIF2α-dependent integrated stress response

We next investigated the molecular mechanism underlying toxicity caused by *C9ORF72* antisense $C_4G_2$ expanded repeats. More than 50 neurological diseases are genetically associated with microsatellite repeat expansions. Repeat expanded RNAs, including CAG, CUG, CCUG, CAGG, and $G_4C_2$, have been shown to activate the double-stranded RNA-dependent protein kinase (PKR) (*Handa et al., 2003*; *Tian et al., 2000*). We hypothesized that *C9ORF72* antisense $C_4G_2$ expanded RNAs can also activate PKR. HEK293T cells expressing in_($C_4G_2$)75 show a significant increase in the level of phosphorylated PKR compared to cells expressing 2 repeats, while the total level of PKR remains unchanged (*Figure 1D and E*). PKR is one of four kinases that are activated during the integrated stress response (ISR), an evolutionarily conserved stress signaling pathway that adjusts cellular biosynthetic capacity according to need (*Martinez et al., 2021*). The four ISR kinases, including PKR, PKR-like ER kinase (PERK), heme-regulated eIF2α kinase (HRI), and general control non-derepressible 2 (GCN2), respond to distinct environmental and physiological stresses by phosphorylating the eukaryotic translation initiation factor eIF2α to cause a temporary shutdown of global protein synthesis and upregulation of specific stress-responsive genes. Accompanying PKR activation, in_($C_4G_2$)75 significantly increases

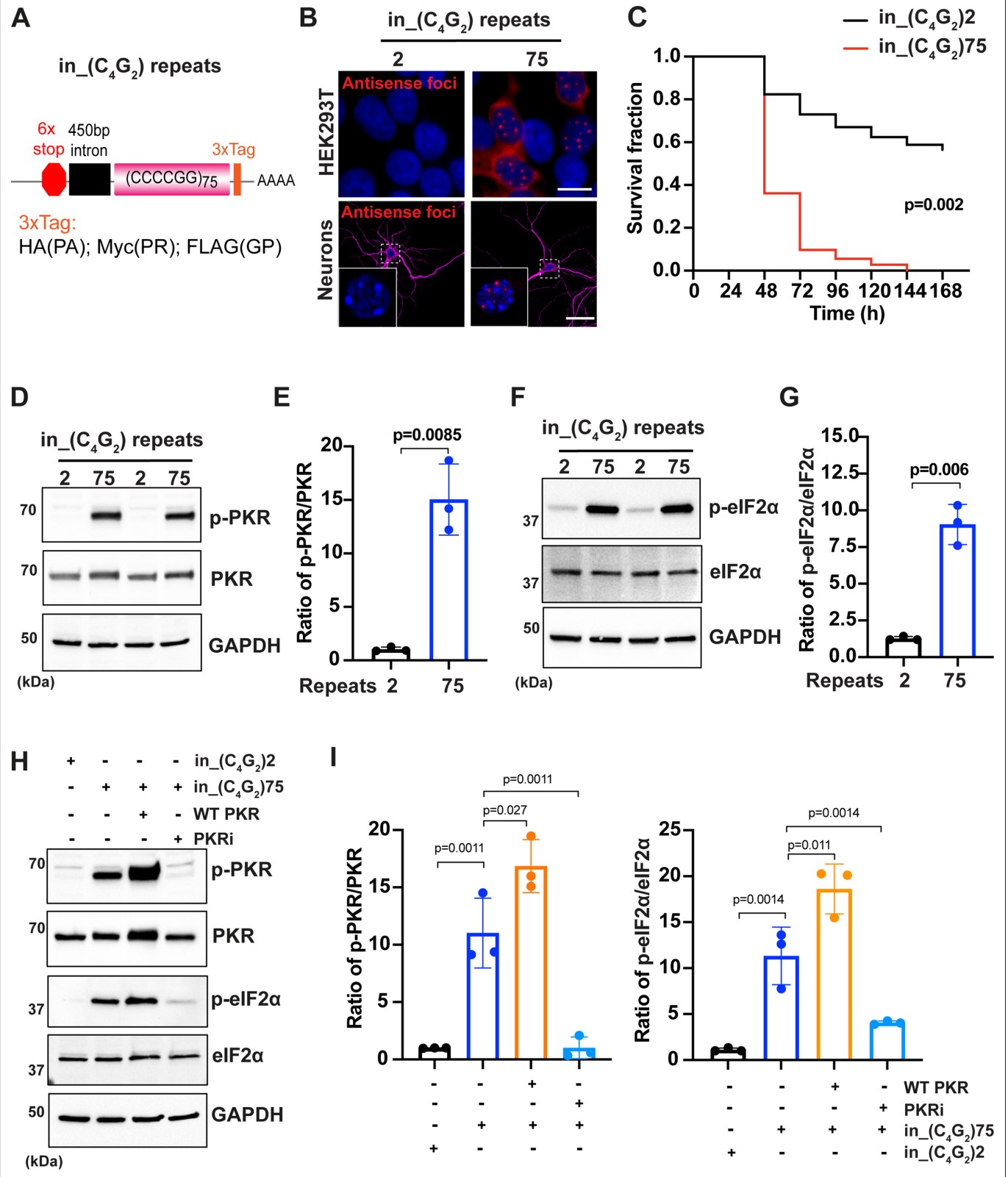

**Figure 1.** *C9ORF72* antisense $C_4G_2$ expanded repeats activate PKR/eIF2α-dependent integrated stress response and cause neuronal toxicity. (**A**) Schematic illustration of the in_$(C_4G_2)$75 repeat construct including 6× stop codons, 450 bp of human intronic sequences at the N-terminus and 3× protein tags at the C- terminus of the repeats to monitor the DPR proteins in each frame. (**B**) Representative images of antisense RNA foci in HEK293T cells and in primary cortical neurons expressing in_$(C_4G_2)$75 detected by RNA FISH. Red, foci; blue, DAPI; magenta, MAP2. (**C**) Kaplan–Meier curves

*Figure 1 continued on next page*

*Figure 1 continued*

showing increased risk of cell death in in_$(C_4G_2)$75-expressing primary cortical neurons compared with neurons expressing 2 repeats. Statistical analyses were performed using Mantel–Cox test (replicated three times with similar results). (**D, E**) Immunoblotting analysis of phosphorylated PKR (p-PKR) and total PKR in HEK293T cells expressing in_$(C_4G_2)$75 or 2 repeats. p-PKR levels were detected using anti-p-PKR (T446) and normalized to total PKR. GAPDH was used as a loading control. Error bars represent SD (n = 3 independent experiments). Statistical analyses were performed using Student's *t*-test. (**F, G**) Immunoblotting analysis of phosphorylated eIF2α (p-eIF2α) and total eIF2α in HEK293T cells expressing in_$(C_4G_2)$75 or 2 repeats. p-eIF2α levels were detected using anti-phosphor eIF2α (Ser51) and normalized to total eIF2α. GAPDH was used as a loading control. Error bars represent SD (n = 3 independent experiments). Statistical analyses were performed using Student's *t*-test. (**H, I**) Immunoblotting analysis of p-PKR and p-eIF2α in HEK293T cells expressing in_$(C_4G_2)$75, with or without co-expression of wild type PKR, or treatment of a PKR inhibitor, C16. Error bars represent SD (n = 3 independent experiments). Statistical analyses were performed using one-way ANOVA with Tukey's post hoc test. Scale bars: 10 μm (neurons), 20 μm (HEK293T).

The online version of this article includes the following source data and figure supplement(s) for figure 1:

**Source data 1.** Original western blot results for *Figure 1D and F*.

**Source data 2.** Original western blot results for *Figure 1H*.

**Figure supplement 1.** *C9ORF72* $C_4G_2$ repeat expanded repeats produce antisense DPR proteins in HEK293T cells and primary neurons.

**Figure supplement 1—source data 1.** Original western blot results for *Figure 1—figure supplement 1B*.

**Figure supplement 2.** *C9ORF72* $C_4G_2$ repeat expanded repeats activate PKR/eIF2α-dependent integrated stress response in SH-SY5Y cells.

**Figure supplement 2—source data 1.** Original western blot results for *Figure 1—figure supplement 2B and C*.

the phosphorylation of eIF2α without affecting its total level (*Figure 1F and G*). in_$(C_4G_2)$75 activates eIF2α mainly by the phosphorylation of PKR as other ISR kinases such as PERK phosphorylation are not altered (*Figure 1—figure supplement 2A*). Consistent with this, overexpressing wild type (WT) PKR further increases the phosphorylation of eIF2α induced by in_$(C_4G_2)$75, whereas treatment with a specific PKR inhibitor C16 reduces the phosphorylation of both PKR and eIF2α to a level comparable to that of cells expressing 2 repeats (*Figure 1H and I*). We further expressed in_$(C_4G_2)$75 in a neuronal cell line SH-SY5Y that is commonly used to study neurodegeneration and observed similar activation of PKR and eIF2α by the antisense $C_4G_2$ expanded repeats (*Figure 1—figure supplement 2B and C*).

To determine whether the activation of PKR/eIF2α leads to a global mRNA translation inhibition, we employed a puromycin-based, nonradioactive method to monitor protein synthesis (*Schmidt et al., 2009*). Puromycin is a structural analog of aminoacyl-tRNA that incorporates into nascent polypeptide chains and prevents elongation. The amount of incorporated puromycin detected by antibodies reflects global translation efficacy. HEK293T cells expressing in_$(C_4G_2)$75 show a significantly reduced amount of incorporated puromycin compared to those expressing 2 repeats (*Figure 2A*). Similarly, neurons expressing 75 antisense $C_4G_2$ repeats, as identified by GP DPR protein accumulation, have a robust global translation inhibition (*Figure 2B and C*). We also observed an abundant accumulation of stress granules in response to stress-induced translation inhibition. Approximately 32% of cells expressing in_$(C_4G_2)$75, identified by the presence of antisense RNA foci, show G3BP1-positive stress granules, and ~55% of foci-positive cells stain for FMRP, another commonly used marker for stress granules (*Figure 2D and E*). These results support that *C9ORF72* antisense $C_4G_2$ expanded repeats activate the PKR/eIF2α-dependent integrated stress response, leading to global translation inhibition and stress granule formation.

## Antisense $C_4G_2$ repeat expanded RNAs activate the PKR/eIF2α pathway independent of DPR proteins

We next determined whether the activation of PKR/eIF2α-dependent integrated stress response is driven by repeat RNA themselves and/or by DPR proteins. We first expressed individual codon-optimized, ATG-driven DPR proteins. Neither PR50, PA50, nor GP80 activates the phosphorylation of eIF2α, suggesting that the activation of the PKR/eIF2α pathway by *C9ORF72* antisense $C_4G_2$ expanded repeats is unlikely due to the DPR proteins produced from RAN translation (*Figure 3—figure supplement 1A and B*). To obtain direct evidence that *C9ORF72* antisense repeat expanded RNAs themselves activate PKR/eIF2α, we used two strategies to reduce/inhibit DPR proteins without affecting the RNA. First, recent studies have shown that *C9ORF72* sense $G_4C_2$ repeat expanded RNAs initiate RAN translation at a near-cognate CUG codon in the intronic region 24 nucleotides upstream of the repeat sequence (*Green et al., 2017*; *Tabet et al., 2018*). We thus hypothesized that

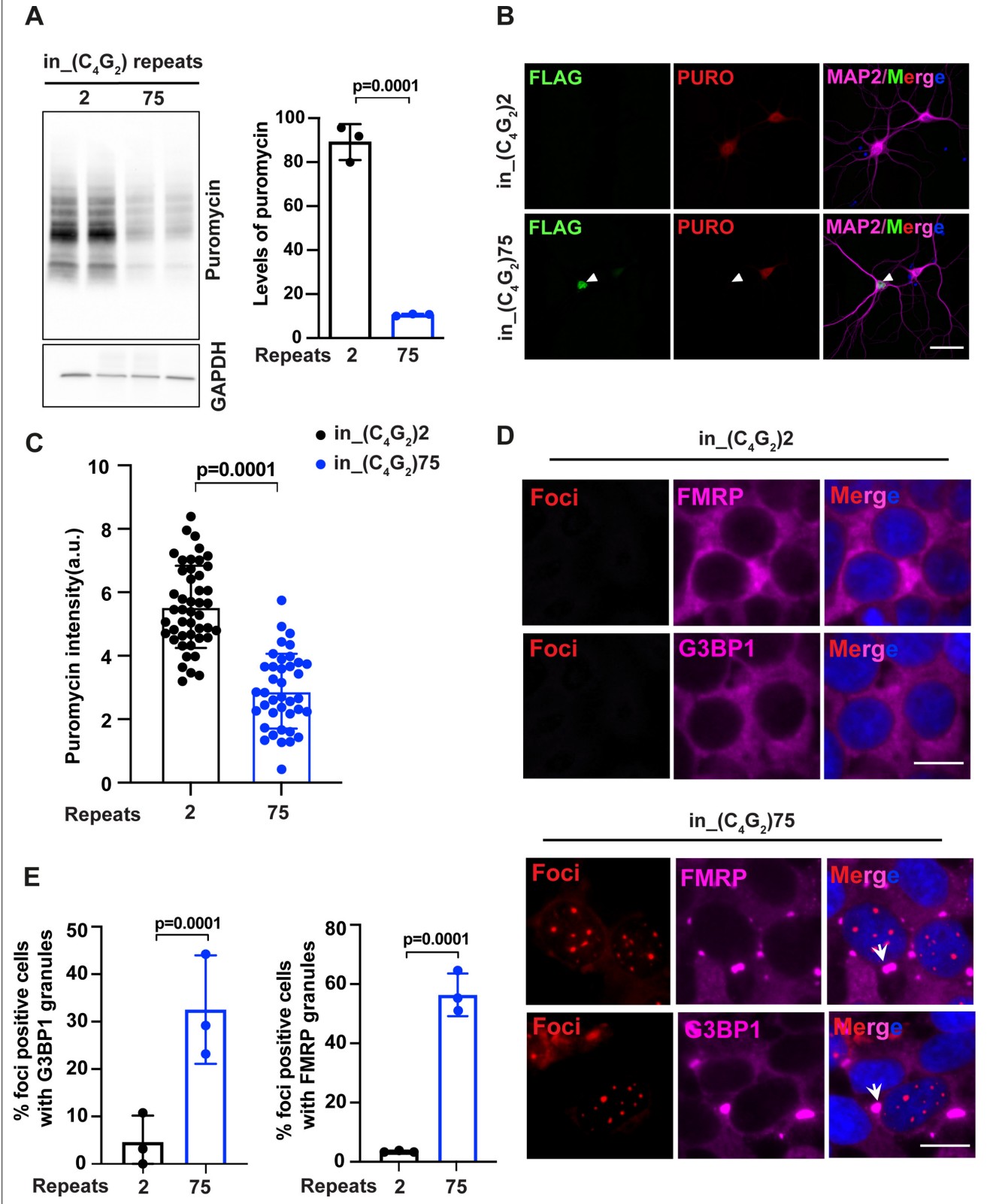

**Figure 2.** *C9ORF72* antisense $C_4G_2$ expanded repeats inhibit global protein synthesis and induce stress granule assembly. (**A**) Immunoblotting of puromycin in HEK293T cells expressing in_$(C_4G_2)$75 or 2 repeats. Cells were incubated with puromycin for 30 min before harvesting. The level of puromycin was normalized to GAPDH. Error bars represent SD (n = 3 independent experiments). Statistical analyses were performed using Student's *t*-test (**B**) Representative images and (**C**) quantification of primary neurons expressing either $(C_4G_2)$75 or 2 repeats stained with anti-puromycin (red),

*Figure 2 continued on next page*

*Figure 2 continued*

anti-FLAG (green), DAPI (blue), and MAP2 (magenta). The puromycin intensity was quantified using ImageJ. Error bars represent SD (n = 40–50 neurons/group; similar results were obtained from two independent experiments). Statistical analyses were performed using Student's *t*-test. (**D**) Representative images of G3BP1 and FMRP staining in HEK293T cells expressing in_(C$_4$G$_2$)75 and in_(C$_4$G$_2$)2 repeats. (**E**) Quantification of antisense foci-positive cells with G3BP1 and FMRP granules. Error bars represent SD (n = 150 cells/condition and three independent experiments). Statistical analyses were performed using Student's *t*-test. Scale bars, 10 μm (neurons), 20 μm (HEK293T).

The online version of this article includes the following source data for figure 2:

**Source data 1.** Original western blot results for *Figure 1A*.

RAN translation from C$_4$G$_2$ antisense repeat expanded RNAs might similarly depend on the intronic sequence at the 5' region. We generated a new construct (C$_4$G$_2$)75 without including the 450 bp human intronic sequence (*Figure 3A*). Supporting our hypothesis, cells expressing (C$_4$G$_2$)75 do not accumulate any detectable GP, PA, or PR DPR proteins, which is strikingly different compared to those expressing in_(C$_4$G$_2$)75 with the 450 bp human intronic sequence (*Figure 3B*). The reduced/abolished DPR proteins by (C$_4$G$_2$)75 are not due to altered RNA expressions since the levels of RNA transcripts and antisense foci are comparable to those of in_(C$_4$G$_2$)75 (*Figure 3C* and *Figure 3—figure supplement 1C*). Second, we obtained a previously reported stop codon-interrupted 108 antisense repeat construct, designated as RNA only (RO) [(C$_4$G$_2$)108RO] (*Figure 3A*). It has been shown that this construct is not RAN translated to produce DPR proteins, while still adopting similar stable conformations as the uninterrupted repeat RNAs (*Moens et al., 2018*). As expected, no detectable antisense DPR proteins are observed in cells expressing (C$_4$G$_2$)108RO, despite an abundant accumulation of antisense foci (*Figure 3—figure supplement 1C and D*). Interestingly, expression of either (C$_4$G$_2$)75 or (C$_4$G$_2$)108RO leads to the robust activation of PKR and eIF2α at a comparable level as seen for in_(C$_4$G$_2$)75 (*Figure 3D and E*). To further validate the above results in a disease-relevant cell type, we made lentivirus to express (C$_4$G$_2$)108RO in primary cortical neurons. We observed the presence of antisense foci in nearly 100% of (C$_4$G$_2$)108RO-expressing neurons (*Figure 3—figure supplement 1E*) together with increased levels of phosphorylated eIF2α (*Figure 3F and G*). Since the available commercial antibodies against mouse phosphorylated PKR do not work in our hands, we designed siRNAs targeting mouse PKR. Reducing PKR in (C$_4$G$_2$)108RO-expressing neurons mitigates the elevated levels of phosphorylated eIF2α (*Figure 3F and G* and *Figure 3—figure supplement 1F and G*). These results support that *C9ORF72* antisense C$_4$G$_2$ repeat expanded RNAs activate the PKR/eIF2α pathway independent of DPR proteins.

## Antisense C$_4$G$_2$ repeat expanded RNAs themselves induce stress granules and lead to neuronal toxicity

Given the conflicting reports of whether *C9ORF72* antisense RNAs themselves are toxic independent of DPR proteins (*Moens et al., 2018*; *Swinnen et al., 2018*; *Tran et al., 2015*), we next focused on (C$_4$G$_2$)108RO, which does not produce DPR proteins but activates the PKR/eIF2α-dependent ISR pathway. We first determined whether the interrupted repeats are sufficient to induce stress granules, which is one of the downstream pathways initiated by phosphorylated eIF2α. We found that FMRP and G3BP1 are mainly diffused in the cytoplasm of cells expressing 2 C$_4$G$_2$ repeats as expected, but they rapidly assemble into stress granules in cells expressing (C$_4$G$_2$)108RO (*Figure 4A–C*). These data suggest that antisense C$_4$G$_2$ repeat expanded RNAs themselves can trigger stress granule formation in the absence of DPR proteins. To determine the role of PKR activation in stress granule formation by (C$_4$G$_2$)108RO, we knocked down *PKR* using siRNAs. siRNAs targeting *PKR* reduce its protein level by 80% compared to control siRNAs (*Figure 4D* and *Figure 3—figure supplement 1H*). Consequently, the phosphorylation of eIF2α by (C$_4$G$_2$)108RO is almost inhibited (*Figure 4D* and *Figure 3—figure supplement 1H*) and the percentage of foci-positive cells with stress granules is significantly reduced (*Figure 4E and F*). Furthermore, a slight but significant increase in the global protein synthesis is also observed (*Figure 3—figure supplement 1I*). Thus, *C9ORF72* antisense C$_4$G$_2$ repeat expanded RNAs themselves induce stress granules by activating PKR/eIF2α.

We further utilized the unbiased longitudinal microscopy assay to determine the risk of death in neurons expressing (C$_4$G$_2$)108RO. Rodent primary cortical neurons were transfected with mApple and (C$_4$G$_2$)108RO or 2 repeats and imaged at 24 hr intervals for 7 d. Neurons expressing (C$_4$G$_2$)108RO show a significant decrease in survival compared to control neurons expressing 2 repeats (*Figure 4G*).

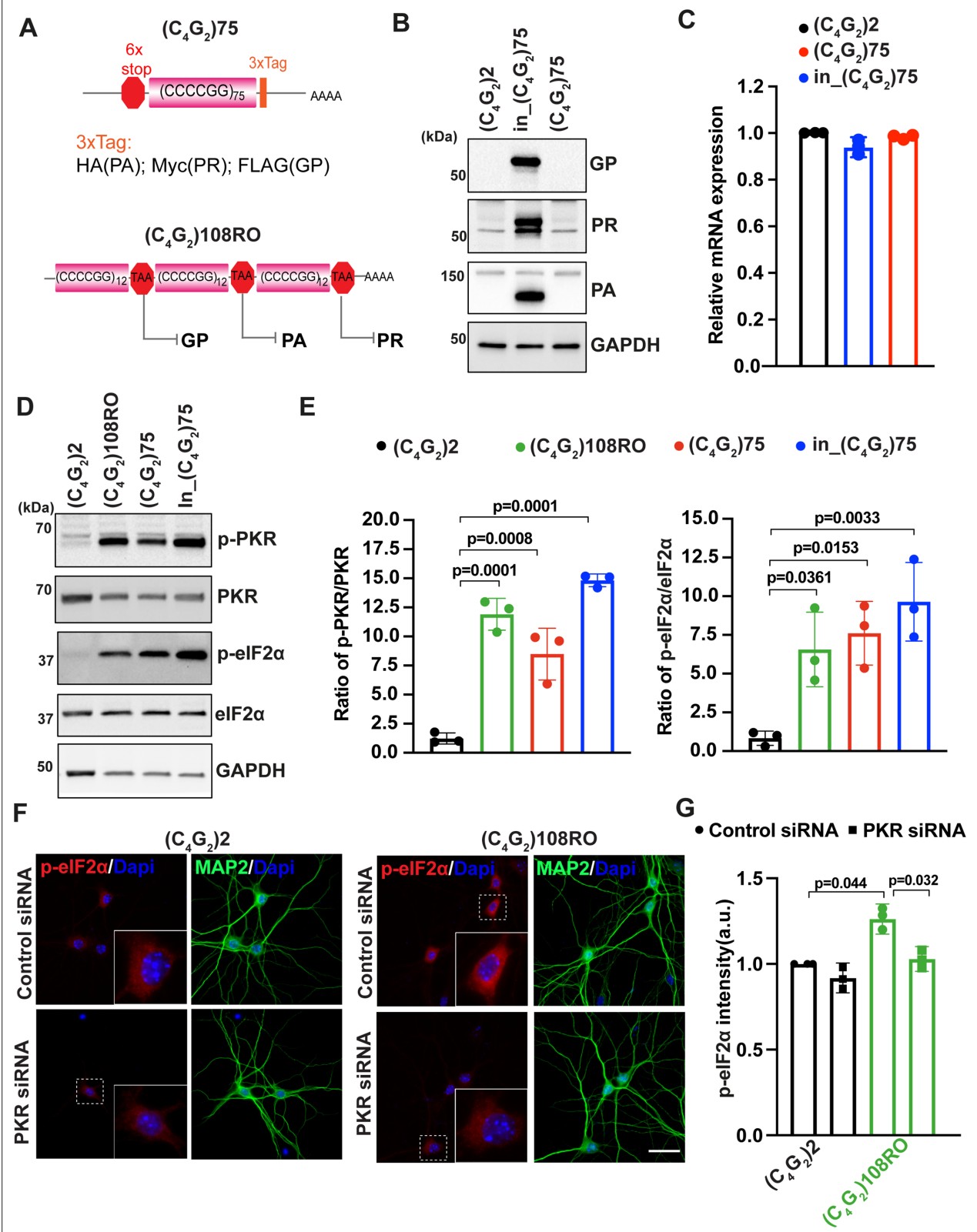

**Figure 3.** Antisense $C_4G_2$ repeat expanded RNAs activate the PKR/eIF2α pathway independent of DPR proteins. (**A**) (Top) Schematic illustration of $(C_4G_2)75$ repeats without the human intronic sequences. 3× protein tags were included at the C- terminus of the repeats to monitor the DPR proteins in each frame. (Bottom) Schematic illustration of antisense $(C_4G_2)108RO$ repeats with stop codons inserted in every 12 repeats to prevent the translation of DPR proteins from all reading frames. (**B**) Immunoblotting of DPR proteins in HEK293T cells expressing in_$(C_4G_2)75$, $(C_4G_2)75$, or 2 repeats. DPR

*Figure 3 continued on next page*

*Figure 3 continued*

protein levels were detected using anti-FLAG (frame with GP), anti-MYC (frame with PR), and anti-HA (frame with PA). GAPDH was used as a loading control. (**C**) mRNA levels were measured by quantitative qPCR in cell expressing in_(C$_4$G$_2$)75, (C$_4$G$_2$)75, or 2 repeats. Error bars represent SD (n = 3). (**D, E**) Immunoblotting of p-PKR and p-eIF2$\alpha$ in HEK293T cells expressing in_(C$_4$G$_2$)75, (C$_4$G$_2$)75, (C$_4$G$_2$)108RO, or 2 repeats. p-PKR (T446) and p-eIF2$\alpha$ (Ser51) were normalized to total PKR and eIF2$\alpha$, respectively. GAPDH was used as a loading control. Error bars represent SD (n = 3 independent experiments). Statistical analyses were performed using one-way ANOVA with Tukey's post hoc test. (**F, G**) Representative images (**F**) and quantitation (**G**) of p-eIF2$\alpha$ in primary neurons expressing antisense (C$_4$G$_2$)108RO or 2 repeats in the presence and absence of PKR siRNA. Scale bars, 10 µm.

The online version of this article includes the following source data and figure supplement(s) for figure 3:

**Source data 1.** Original western blot results for *Figure 3B*.

**Source data 2.** Original western blot results for *Figure 3D*.

**Figure supplement 1.** Antisense DPR proteins do not activate PKR/eIF2$\alpha$-dependent integrated stress response.

**Figure supplement 1—source data 1.** Original western blot results for *Figure 3—figure supplement 1D*.

**Figure supplement 1—source data 2.** Original western blot results for *Figure 3—figure supplement 1F*.

Importantly, knockdown of PKR partially rescues the (C$_4$G$_2$)108RO-mediated toxicity (*Figure 4G*). These data suggest that *C9ORF72* antisense repeat expanded RNAs themselves are neurotoxic via PKR activation.

## Increased levels of phosphorylated PKR and eIF2$\alpha$ in C9FTD/ALS patients

To study disease relevance, we next determined the levels of phosphorylated PKR and eIF2$\alpha$ in C9FTD/ALS patient postmortem tissues and in patient-derived iPSCs motor neurons. Immunohistochemistry staining showed that the level of phosphorylated PKR is increased in the frontal cortex, especially in the large pyramidal neurons, of patients carrying *C9ORF72* repeat expansions compared to age-matched non-disease controls (*Figure 5A*). This data is consistent with two recent studies showing increased PKR phosphorylation in *C9ORF72* patients (*Rodriguez et al., 2021*; *Zu et al., 2020*). In addition, the level of phosphorylated eIF2$\alpha$ after normalizing to the total eIF2$\alpha$ level is also significantly increased, despite the heterogeneity of eIF2$\alpha$ protein levels in patients (*Figure 5B and C*, *Table 1*). These results suggest that the PKR/eIF2$\alpha$ pathway is activated in C9FTD/ALS patients. We also obtained protein extracts from two lines of *C9ORF72* patient-derived motor neurons and their isogenic controls (*Braems et al., 2022*). However, we did not observe any differences in the levels of either phosphorylated PKR or phosphorylated eIF2$\alpha$ of 38-day-old neurons (*Figure 5—figure supplement 1A and B*), suggesting that PKR/ eIF2$\alpha$-dependent ISR pathway possibly gets activated in *C9ORF72* patients in a later stage of disease pathogenesis.

## Sense G$_4$C$_2$ repeat expanded RNAs cannot activate the PKR/eIF2$\alpha$ pathway

Recently, Zu et al. showed that expressing a construct containing (G$_4$C$_2$)120 also activates PKR and increases DPR protein translation in HEK293T cells (*Zu et al., 2020*). However, it is not known whether this activation is due to sense or antisense transcripts. Therefore, we generated a sense repeat construct (G$_4$C$_2$)75 that has a similar repeat size as our antisense construct (*Figure 6—figure supplement 1A*). Consistent with the earlier findings by Zu et al., expression of sense (G$_4$C$_2$)75 in HEK293T cells significantly increases phosphorylation of both PKR and eIF2$\alpha$ (*Figure 6A* and *Figure 6—figure supplement 1B*). However, we detected abundant accumulation of both sense and antisense RNA foci in cells expressing sense (G$_4$C$_2$)75 but not in those expressing 2 repeats (*Figure 6—figure supplement 1C*). Interestingly, we also detected accumulation of sense foci in cells expressing antisense (C$_4$G$_2$) expanded repeats in such model system (*Figure 6—figure supplement 1C*). We therefore aimed to determine the relative contribution of sense G$_4$C$_2$ and antisense C$_4$G$_2$ repeat expanded RNAs in activating the PKR/eIF2$\alpha$ pathway by focusing the expanded (G$_4$C$_2$)75 repeats which resemble C9FTD/ALS patients. We first used previously published ASOs that specifically degrade sense G$_4$C$_2$ RNAs (*Jiang et al., 2016*). As expected, ASOs targeting sense G$_4$C$_2$ repeat expanded RNAs significantly reduce the accumulation of sense RNA foci but have little effect on antisense RNA foci (*Figure 6B* and *Figure 6—figure supplement 1D*). However, reducing sense G$_4$C$_2$ RNA transcripts/foci does not alter the activation of PKR/eIF2$\alpha$ by (G$_4$C$_2$)75 (*Figure 6C and D*). We next designed two ASOs specifically

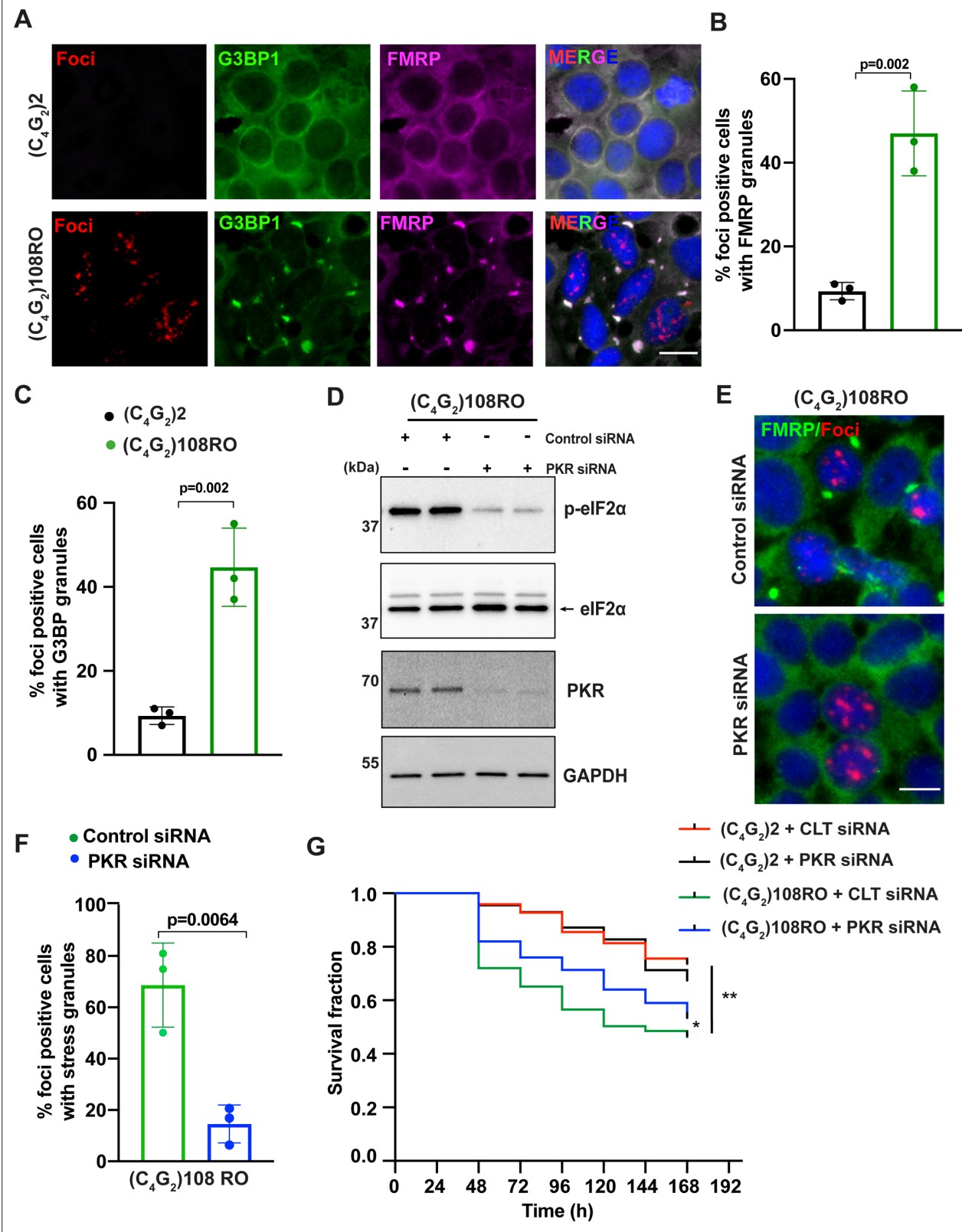

**Figure 4.** Antisense C$_4$G$_2$ repeat expanded RNAs themselves induce stress granules and lead to neuronal toxicity. (**A**) Representative images of FMRP and G3BP1 staining in HEK293T cells expressing (C$_4$G$_2$)108RO or 2 repeats. (**B, C**) Quantification of antisense foci-positive cells with FMRP and G3BP1 granules. Error bars represent SD (n = 180 cells/condition and three independent experiments). Statistical analyses were performed using Student's *t*-test. (**D**) Immunoblotting of PKR and p-eIF2α (Ser51) in HEK293T cells expressing (C$_4$G$_2$)108RO together with control or PKR siRNA. GAPDH was used

*Figure 4 continued on next page*

*Figure 4 continued*

as a loading control. (**E**) Representative images of FMRP staining in HEK293T cells expressing $(C_4G_2)108$ repeats together with either control or PKR siRNA. (**F**) Quantification of antisense foci-positive cells with FMRP granules. Error bars represent SD (n = 150 cells/condition and three independent experiments). (**F**) Kaplan–Meier curves showing the risk of cell death in $(C_4G_2)108RO$-expressing neurons compared with 2 repeats in the presence and absence of PKR siRNA (replicated three times with similar results). Statistical analyses were performed using Mantel–Cox test (*p<0.05, **p<0.01). Scale bars, 20 μm.

The online version of this article includes the following source data for figure 4:

**Source data 1.** Original western blot results for *Figure 4D*.

targeting antisense repeat RNAs. Both ASO1 and ASO2 targeting antisense $C_4G_2$ repeat expanded RNAs significantly reduce the abundance of antisense RNA foci without affecting sense RNA foci (*Figure 6B* and *Figure 6—figure supplement 1E*). Consequently, both ASOs significantly inhibit the activation of PKR and eIF2α by $(G_4C_2)75$ (*Figure 6E and F*). We further obtained an interrupted sense $(G_4C_2)108RO$ construct that has the similar repeat size as the antisense $(C_4G_2)108RO$ (*Moens et al., 2018*). Expression of sense $(G_4C_2)108RO$ also significantly increases the phosphorylation of PKR and eIF2α, which is mitigated by ASOs targeting antisense RNAs, but not sense RNAs (*Figure 6—figure supplement 1F*). These results suggest that antisense $C_4G_2$, but not sense $G_4C_2$, repeat expanded RNAs (either pure repeats or interrupted) activate the PKR/eIF2α-mediated ISR pathway. It has been shown previously that expressing sense repeats can induce stress granules (*Green et al., 2017*), but the underlying molecular mechanisms are not explored. We observed a significant increase in the accumulation of FMRP and G3BP1 granules in $(G_4C_2)75$ repeats expressing cells compared to those expressing the control 2 repeats. Interestingly, such aberrant accumulation of stress granules is drastically reduced after treatment with ASOs targeting the antisense $G_4C_2$ repeat RNAs (*Figure 6G and H*). Thus, antisense $C_4G_2$, but not sense $G_4C_2$, repeat expanded RNAs activate the PKR/eIF2α pathway, leading to toxicity.

## Reduction of PKR mitigates antisense $C_4G_2$, but not sense $G_4C_2$, RNA-mediated toxicity in zebrafish

To validate our findings in an in vivo model, we performed a morpholino-mediated knockdown of the zebrafish ortholog of human PKR (i.e. Eif2ak2) in zebrafish expressing *C9ORF72* repeat expanded RNAs. We previously demonstrated that micro-injection of RNAs containing either sense $(G_4C_2)70$ or antisense $(C_4G_2)70$ repeats results in motor axonopathy including reduced axonal length and aberrant branching in zebrafish embryos (*Swinnen et al., 2018*). As no DPR proteins are detected, this phenotype is specifically caused by sense or antisense RNA themselves. We first designed a splice-blocking morpholino (SB-MO) targeting the exon 3–intron 3 junction of zebrafish *eif2ak2*. SB-MO caused the retention of intron 3 in a dose–response manner, resulting in a reduction of the WT *eif2ak2* transcripts (*Figure 7A* and *Figure 7—figure supplement 1A*). Co-injection of antisense $(C_4G_2)70$ RNAs with *eif2ak2* SB-MO significantly mitigates the abnormalities of axonal length and branching compared to zebrafish injected with $(C_4G_2)70$ RNAs and a control morpholino (*Figure 7B–D*). Consistent with this observation, Eif2ak2 reduction with a translation-blocking morpholino increases axonal length (*Figure 7A* and *Figure 7—figure supplement 1B and C*) and significantly ameliorates the abnormal branching phenotype caused by antisense $(C_4G_2)70$ RNAs. On the other hand, no protective effects were observed when co-injecting sense $(G_4C_2)70$ RNAs with *eif2ak2* SB-MO (*Figure 7E and F*). Overall, these data indicate that PKR is an important player contributing to toxicity induced by antisense $C_4G_2$ but not sense $G_4C_2$ repeat expanded RNAs.

## Discussion

It is generally accepted in the field that gain of toxicity from the bidirectionally transcribed repeat expanded RNAs plays important roles in FTD/ALS caused by *C9ORF72* repeat expansions (*Jiang and Ravits, 2019*). However, the relative contributions of potentially toxic species, including sense and antisense RNAs themselves, RNA foci, and DPR proteins, are largely debated. Our study shows for the first time that *C9ORF72* antisense $C_4G_2$ repeat expanded RNAs activate PKR/eIF2α-dependent integrated stress response and lead to neurotoxicity independent of DPR proteins in model systems (*Figure 8*). Although most C9FTD/ALS patients carry significantly longer $G_4G_2$ repeats and

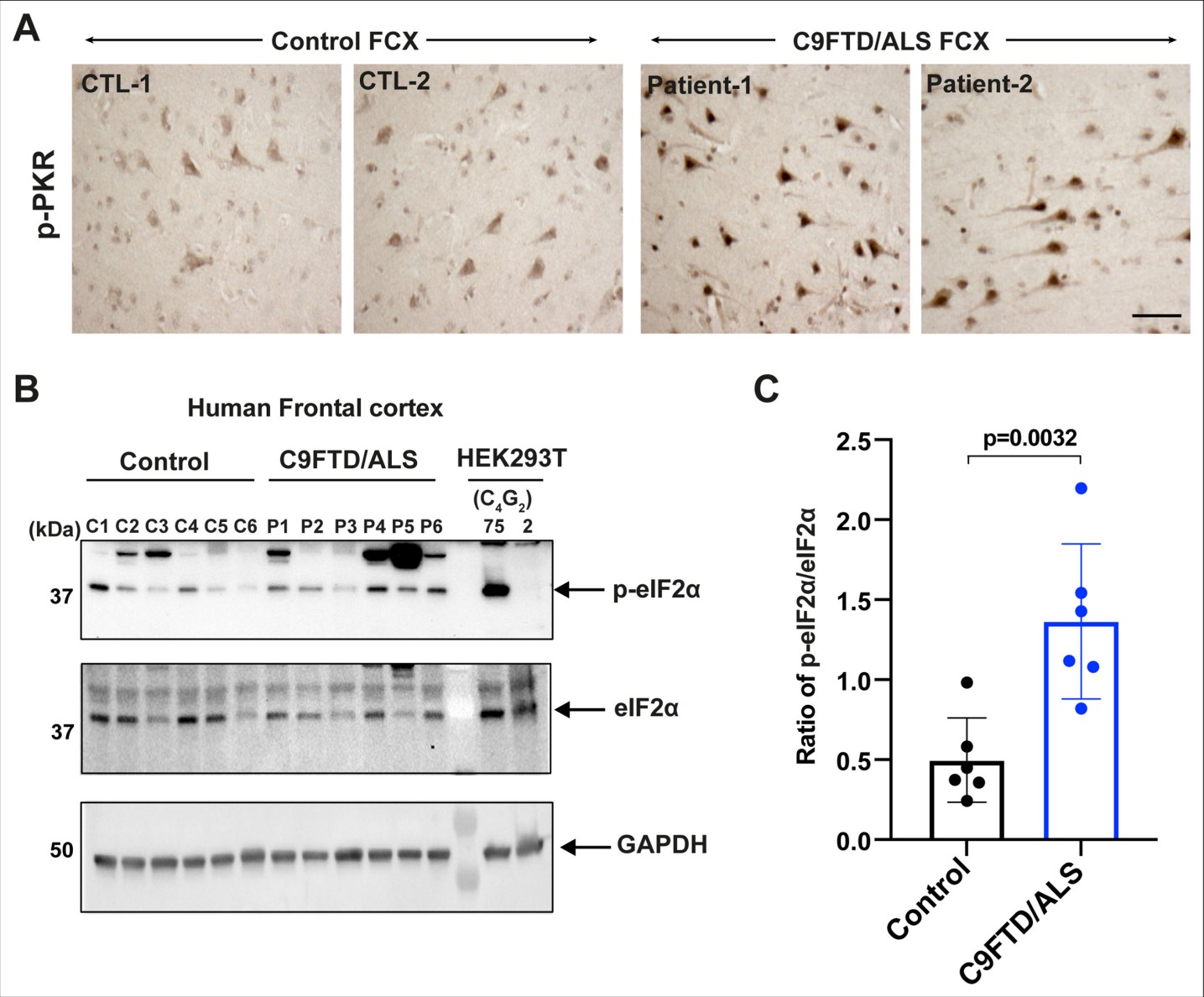

**Figure 5.** Increased levels of phosphorylated PKR and eIF2α in C9FTD/ALS patients. (**A**) Representative immunohistochemistry images of phosphorylated PKR staining in control and C9FTD/ALS patient's frontal cortex (FCX) using anti-p-PKR (T446) (n = 4 per genotype). (**B, C**) Immunoblotting of p-eIF2α in proteins extracted from control (C1–C6) and C9FTD/ALS patient's frontal cortex (P1–P6). p-eIF2α (Ser51) was normalized to total eIF2α. GAPDH was used as a loading control. Error bars represent SD (control n = 6 and C9FTD/ALS n = 6). Statistical analyses were performed using unpaired Student's t-test. Scale bars, 10 μm.

The online version of this article includes the following source data and figure supplement(s) for figure 5:

**Source data 1.** Original western blot results for *Figure 5B*.

**Figure supplement 1.** Phosphorylated PKR and eIF2α levels are unchanged in C9FTD/ALS iPSC-derived motor neurons.

**Figure supplement 1—source data 1.** Original western blot results for *Figure 5—figure supplement 1A*.

may produce antisense C₄G₂ repeat expanded RNAs of different sizes and expression levels, we also detected increased activation of PKR/eIF2α in patient frontal cortex. Consistent with our observations, increased phosphorylation of PKR has also been reported in BAC transgenic mice expressing 500 G₄C₂ repeats and in *C9ORF72* patients by two other studies (*Rodriguez et al., 2021*; *Zu et al., 2020*). Importantly, using strand-specific ASOs we showed the previously reported increased phosphorylation of PKR/eIF2α and stress granules when expressing G₄C₂ repeats are mainly driven by

**Table 1.** List of controls (C1–C6) and C9FTD/ALS patients (P1–P6) postmortem tissues used in this study.

| ID | Primary neuropathological diagnosis | Age at onset (years) | Age at death (years) | Sex |
|----|--------------------------------------|----------------------|----------------------|-----|
| C1 | Control | - | 53 | F |
| C2 | Control | - | 77 | NA |
| C3 | Control | - | 43 | F |
| C4 | Control | - | 57 | M |
| C5 | Control | - | 72 | F |
| C6 | Control | - | 57 | F |
| P1 | FTLD (C9 expansion) | 58 | 71 | M |
| P2 | ALS (C9 expansion) | 54 | 57 | F |
| P3 | FTLD (C9 expansion) | 57 | 66 | M |
| P4 | FTLD (C9 expansion) | 58 | 67 | M |
| P5 | FTLD (C9 expansion) | 62 | 66 | M |
| P6 | ALS (C9 expansion) | - | 69 | NA |

ALS: amyotrophic lateral sclerosis.

the antisense transcripts (*Figures 6 and 7*), highlighting the significance of antisense $C_4G_2$ repeat expanded RNAs in disease pathogenesis.

Several studies argue against the toxicity from either sense $G_4C_2$ or antisense $C_4G_2$ repeat expanded RNAs and associated RNA foci. In one study, *Drosophila* expressing intronic $(G_4C_2)160$ repeats show abundant sense RNA foci in the nucleus but have little DPR proteins and no neurodegeneration, suggesting that sense RNA foci is insufficient to cause toxicity in this model (*Tran et al., 2015*). Two other elegant studies generated *Drosophila* expressing interrupted $G_4C_2$ repeats by inserting stop codons every 12 repeats in all reading frames to prevent RAN translation. These *Drosophila* do not show any toxicity, whereas those expressing pure $G_4C_2$ repeats of similar sizes do despite comparable sense RNA foci accumulation. Similarly, *Drosophila* expressing interrupted antisense repeat expanded RNAs do not show any deficits, suggesting that antisense $C_4G_2$ RNAs are not toxic in this model system (*Mizielinska et al., 2014*; *Moens et al., 2018*). However, expressing the same antisense interrupted repeat construct $(C_4G_2)108RO$ causes motor axonopathy in zebrafish (*Swinnen et al., 2018*). Consistent with this, we show that $(C_4G_2)108RO$ is toxic to primary cortical neurons, which is partially rescued by the reduction of PKR (*Figure 4G*). The PKR-dependent toxicity is also observed in zebrafish expressing antisense $(C_4G_2)70$ RNAs (*Figure 7B–D* and *Figure 7—figure supplement 1B and C*). It is interesting to note that PKR, constitutively and ubiquitously expressed in vertebrate cells including zebrafish, is not found in plants, fungi, protists, or invertebrates such as *Drosophila* (*Martinez et al., 2021*). This may partially explain the difference in toxicities caused by the antisense $C_4G_2$ repeat expanded RNAs in *Drosophila*, zebrafish, and primary neurons.

The contribution of antisense repeat expanded RNAs to C9FTD/ALS pathogenesis is understudied, although overexpression of PR DPR proteins as those RAN translated from the antisense RNAs has been shown to be toxic in various model systems (*Jovičić et al., 2015*; *Wen et al., 2014*; *Zhang et al., 2019*). Aggregates of PR DPR proteins are rare in C9FTD/ALS postmortem tissues, whereas antisense RNA foci are as abundant as sense RNA foci in multiple CNS regions despite the scarcity of antisense RNA transcripts. One possibility of such discrepancy between RNA transcripts and foci levels is that antisense RNA foci are extraordinarily stable and rarely turn over once formed along the life span of patients. Several neuropathological studies have attempted to correlate the abundance and distribution of antisense RNA foci with C9FTD/ALS clinical features. *Mizielinska et al., 2013* showed that patients with more antisense RNA foci tend to have an earlier age of symptom onset and, more intriguingly, antisense RNA foci are shown to be associated with nucleoli and mislocalization of TDP-43 in two different studies (*Aladesuyi Arogundade et al., 2019*; *Cooper-Knock et al., 2015*).

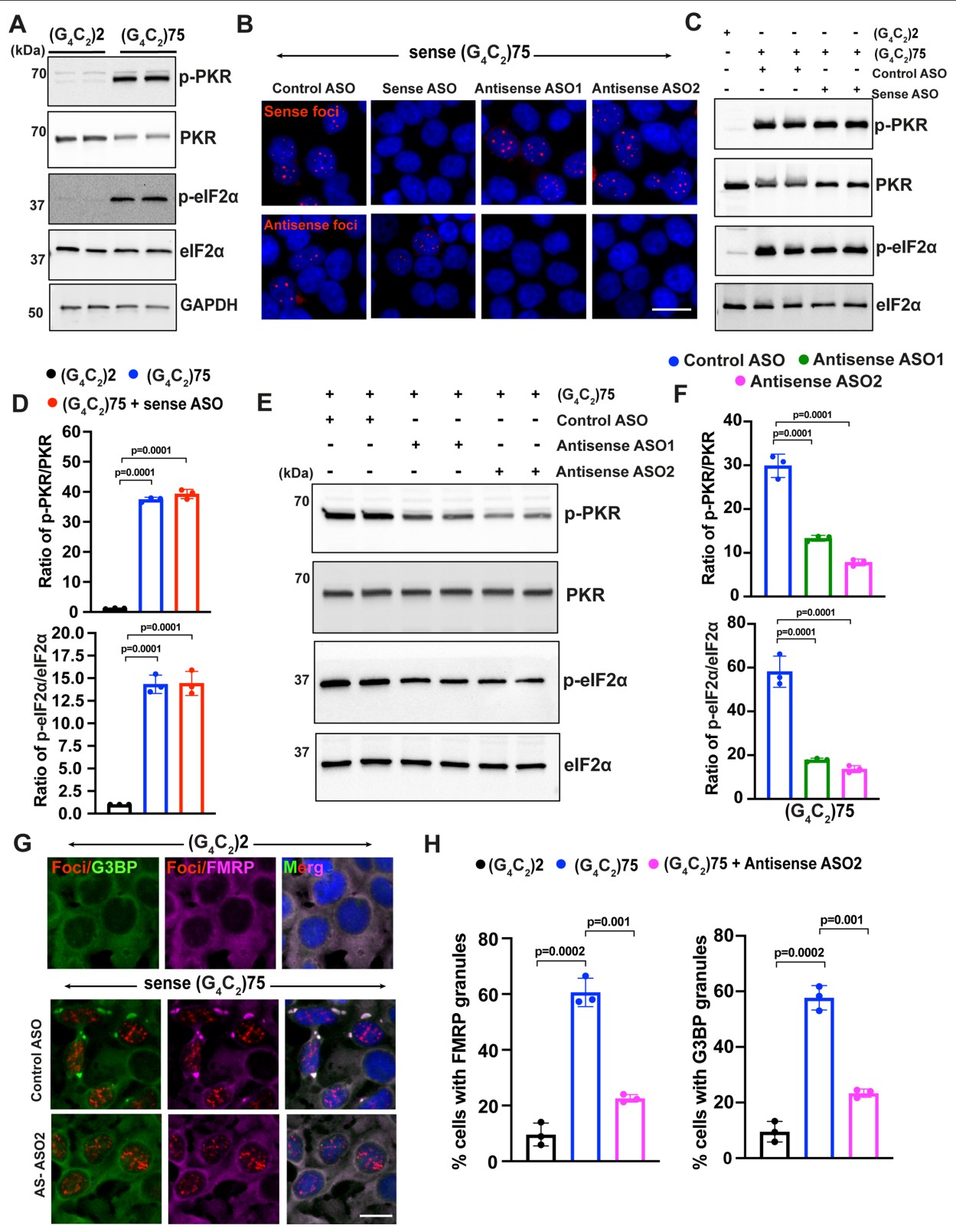

**Figure 6.** Sense $G_4C_2$ repeat expanded RNAs cannot activate the PKR/eIF2α pathway. (**A**) Immunoblotting of p-PKR and p-eIF2α in HEK293T cells expressing $(G_4C_2)75$ or $(G_4C_2)2$ repeats. (**B**) Representative images of sense and antisense RNA foci in HEK293T expressing $(G_4C_2)75$ repeats together with either control antisense oligonucleotides (ASOs), ASOs targeting sense $G_4C_2$RNAs or ASOs targeting antisense $C_4G_2$ repeat expanded RNAs. Foci were detected by RNA FISH. Red, foci; blue, DAPI. (**C**) Immunoblotting and (**D**) quantification of p-PKR and p-eIF2α in HEK293T cells expressing

*Figure 6 continued on next page*

*Figure 6 continued*

$(G_4C_2)75$ or $(G_4C_2)2$ repeats together with either control ASO or ASOs targeting sense $G_4C_2$ repeat expanded RNAs. Phosphorylated PKR levels were detected using anti-p-PKR (phosphor T446) and normalized to total PKR. Phosphorylated eIF2α levels were detected using anti-peIF2α (Ser51) and normalized to total eIF2α. GAPDH was used as a loading control. Error bars represent SD (n = 3 independent experiments). Statistical analyses were performed using unpaired Student's *t*-test. (**E, F**) Immunoblotting of p-PKR and p-eIF2α in HEK293T cells expressing $(G_4C_2)75$ or $(G_4C_2)2$ repeats together with either control ASO or ASOs targeting antisense $G_4C_2$ repeat expanded RNAs. P-PKR (T446) and p-eIF2α (Ser51) were normalized to total PKR and eIF2α, respectively. GAPDH was used as a loading control. Error bars represent SD (n = 3 independent experiments). Statistical analyses were performed using one-way ANOVA with Tukey's post hoc test. (**G**) Representative images and (**H**) quantification of FMRP and G3BP1 staining in HEK293T cells expressing $(G_4C_2)75$ or $(G_4C_2)2$ repeats in the presence and absence of antisense ASO2. Error bars represent SD (n = 150 cells/condition and three independent experiments). Statistical analyses were performed using one-way ANOVA with Tukey's post hoc test. Scale bars, 20 μm.

The online version of this article includes the following source data and figure supplement(s) for figure 6:

**Source data 1.** Original western blot results for *Figure 6A*.

**Source data 2.** Original western blot results for *Figure 6C*.

**Source data 3.** Original western blot results for *Figure 6E*.

**Figure supplement 1.** Sense $G_4C_2$ repeat expanded RNAs cannot activate the PKR/eIF2α pathway.

**Figure supplement 1—source data 1.** Original western blot results for *Figure 6—figure supplement 1F*.

These results highlight the disease relevance of antisense repeat expanded RNAs in C9FTD/ALS and the significance of our work. Our study, however, does not differentiate antisense RNA foci from RNAs themselves since it is technically challenging given that RNA foci inevitably form with the expression of repeat expanded RNAs. The proposed mechanisms of RNA foci-mediated toxicity are via sequestration of critical RBPs. Several RBPs have been proposed to interact with and/or are sequestered into sense RNA foci, yet those interacting with antisense RNA foci have not been well characterized and are worth exploring especially in correlation with PKR/eIF2α activation.

How is PKR specifically activated by *C9ORF72* antisense but not sense repeat expanded RNAs? PKR is a stress sensor first identified as a kinase responding to viral infections by directly binding to viral double-stranded RNAs (dsRNAs) (*Martinez et al., 2021*). Several disease-relevant repeats expanded RNAs, such as CUG and CGG, have been shown to form stable hairpins and directly bind to PKR, leading to its activation (*Handa et al., 2003*; *Tian et al., 2000*). It has been shown that antisense $C_4G_2$ DNAs form i-motifs consisting of two parallel duplexes in a head to tail orientation as well as protonated hairpins under near-physiological conditions, thus posing a possibility of direct activation of PKR by antisense RNA (*Kovanda et al., 2015*). In contrast, sense $G_4C_2$ RNAs exist in an equilibrium between two folded states, hairpin and G-quadruplex (*Wang et al., 2019*), and tend to form more stable unimolecular and multimolecular G-quadruplexes (*Fratta et al., 2012*; *Reddy et al., 2013*). Indeed, *Rodriguez et al., 2021* showed dsRNAs, derived at least in part from G4C2 repeats, colocalize with phosphorylated TDP-43 in the cerebellum and frontal cortex of C9FTD/ALS patients. In addition to *C9ORF72* antisense repeat expanded RNAs, short and long interspersed retrotransposable elements (SINEs and LINEs) and endogenous retroviruses (ERVs) represent other main sources of endogenous dsRNAs (*Sadeq et al., 2021*). In *C9ORF72* patients, transcripts from multiple classes of repetitive elements are significantly elevated (*Prudencio et al., 2017*). Other PKR activators include cellular stresses such as oxidative stress, intracellular calcium increase or ER stress, as well as interferon-gamma (IFNγ), tumor necrosis factor α (TNFα), heparin, and platelet-derived growth factor (*Martinez et al., 2021*). Whether *C9ORF72* antisense $C_4G_2$ expanded RNAs activate PKR directly or indirectly via increased transcription of RNAs with repetitive elements or other PKR activators warrants additional studies.

Our study shows that *C9ORF72* antisense $C_4G_2$ expanded repeat RNAs promote robust global translation inhibition and stress granule formation independent of DPR proteins via the activation of the PKR/eIF2α pathway. Stress granules are dynamic structures that form and disperse rapidly with acute stress. However, chronic stress during aging or under pathological conditions leads to altered stress granule dynamics and persistent stress granules, which have been implicated in the aggregation of RBPs such as TDP-43 and contribute to the pathogenesis of FTD and ALS (*Li et al., 2013*). Previous studies have shown that expressing sense $(G_4C_2)$ repeats can activate the PKR/eIF2a pathway (*Zu et al., 2020*) and induce stress granules (*Green et al., 2017*), which is replicated in our study (*Figure 6*). Interestingly, we detected abundant accumulation of both sense and antisense RNA foci

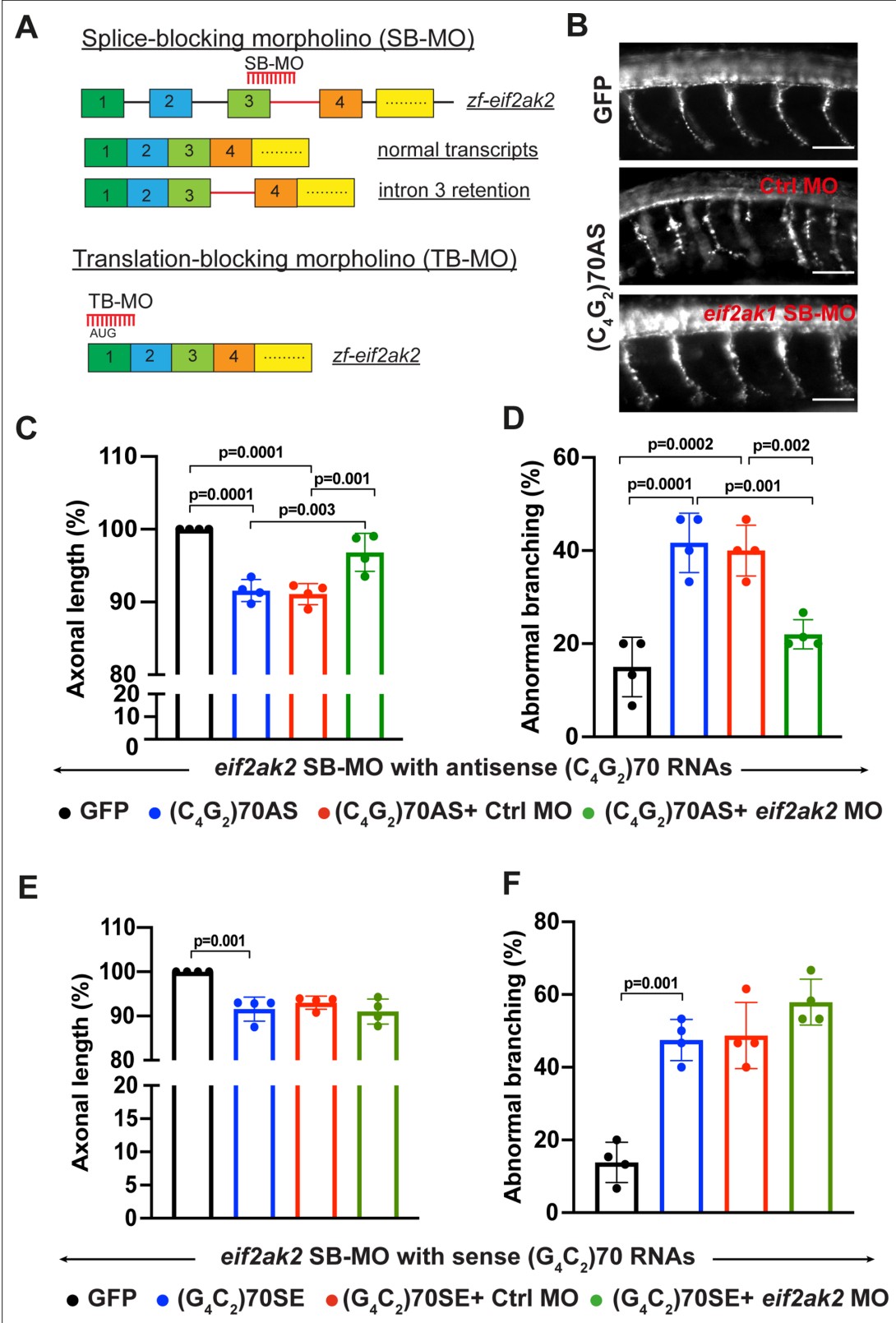

**Figure 7.** Reduction of PKR mitigates antisense $C_4G_2$, but not sense $G_4C_2$, RNA- mediated toxicity in zebrafish. (**A**) Schematic illustration of splice-blocking morpholino (SB-MO) targeting the exon 3/intron 3 junction in zebrafish *eif2ak2* pre-mRNA. The correctly spliced (wild type transcripts) and intron 3 retained transcripts are shown. The translation-blocking morpholino targets the AUG start codon in post-spliced *eif2ak2* mRNA. (**B**) SV2 immunostaining of motor axons in 30 hpf zebrafish embryos injected with GFP control mRNA and antisense $(C_4G_2)70$ RNAs. Scale bar = 50 µm. (**C, D**)

*Figure 7 continued on next page*

*Figure 7 continued*

Effects of SB-MO on axonal length (**C**) and branching (**D**) of zebrafish expressing antisense (C$_4$G$_2$)70 RNAs. Embryos were injected with 0.844 µM GFP mRNA, 0.844 µM antisense (C$_4$G$_2$)70 RNAs, 0.844 µM antisense (C$_4$G$_2$)70 RNAs plus 0.25 mM control morpholino, or 0.844 µM antisense (C$_4$G$_2$)70 RNAs plus 0.25 mM *eif2ak2* morpholino. Error bars represent SD (n = 4 independent experiments). Statistical analyses were performed using one-way ANOVA with Tukey's post hoc test. (**E, F**) Effects of SB-MO on axonal length and branching of zebrafish expressing sense (G$_4$C$_2$)70 RNAs. Embryos were injected with 0.844 µM GFP mRNA, 0.844 µc sense (G$_4$C$_2$)70 RNAs, 0.844 µM sense (G$_4$C$_2$)70 RNAs plus 0.25 mM control morpholino or 0.844 µM sense (G$_4$C$_2$)70 RNAs plus 0.25 mM *eif2ak2* morpholino. Error bars represent SD (n = 4 independent experiments). Statistical analyses were performed using one-way ANOVA with Tukey's post hoc test.

The online version of this article includes the following source data and figure supplement(s) for figure 7:

**Figure supplement 1.** Translation-blocking morpholino of PKR mitigates antisense C$_4$G$_2$ repeat RNA-induced axonopathy in zebrafish.

**Figure supplement 1—source data 1.** Original agarose gel results for *Figure 7—figure supplement 1A*.

in cells expressing sense (G$_4$C$_2$) expanded repeats. How such bidirectional transcription for *C9ORF72*-associated expanded repeats initiates is not known. Of note, we also detected sense foci accumulation in HEK293T cells expressing antisense (C$_4$G$_2$) expanded repeats. Nevertheless, using strand-specific ASOs to selectively degrade sense/antisense RNAs (*Figure 6*) or by directly injecting sense/antisense RNAs in zebrafish (*Figure 7*), our data clearly show that antisense C$_4$G$_2$, but not sense G$_4$C$_2$, repeat expanded RNAs activate the PKR/eIF2α pathway, leading to toxicity. It has been suggested the RAN translation of DPR proteins is increased explicitly by the integrated stress response via eIF2α phosphorylation (*Cheng et al., 2018*; *Green et al., 2017*) and several DPR proteins such as GR, PR, and GA are toxic (*Boeynaems et al., 2016*; *Choi et al., 2019*; *Freibaum et al., 2015*; *Hao et al., 2019*; *Kanekura et al., 2016*; *Lee et al., 2016*; *May et al., 2014*; *Mizielinska et al., 2014*; *Schludi et al., 2017*; *Tao et al., 2015*; *Wen et al., 2014*; *Yamakawa et al., 2015*; *Yang et al., 2015*; *Zhang et al., 2018*; *Zhang et al., 2016*; *Zhang et al., 2019*; *Zhang et al., 2014*; *Zu et al., 2011*). Thus, activation of PKR/eIF2α by *C9ORF72* antisense repeat expanded RNAs may lead to additional accumulation of DPR proteins and toxicity. Future studies will determine the relative contributions and interplay between repeat containing RNAs from either direction and DPRs proteins to disease pathogenesis.

From the therapeutic development point of view, several approaches have been explored to mitigate the gain of toxicity from *C9ORF72* repeat expanded RNAs (*Jiang and Ravits, 2019*). Our previous work with ASOs specifically targeting *C9ORF72* sense G$_4$C$_2$ repeat expanded RNAs showed great promise in a preclinical mouse model expressing 450 G$_4$C$_2$ repeats (*Jiang et al., 2016*). Unfortunately, there was a recent setback in a clinical trial using these ASOs to treat C9ALS patients (Mar 2022, news release by Biogen Inc). Although many confounding reasons may cause drug failures, further understanding of disease mechanisms is required to develop successful therapies for C9FTD/ALS. On this note, Zu et al. recently showed that metformin, an FDA-approved drug widely used for treating type 2 diabetes, inhibits PKR activation, reduces DPR proteins RAN translated from the sense strand, and improves behavioral and pathological deficits in BAC transgenic mice expressing G$_4$C$_2$ repeats (*Zu et al., 2020*). Our study now suggests that the activation of PKR in these BAC transgenic mice and in *C9ORF72* patients might result from the antisense repeat expanded RNAs. Future therapies targeting *C9ORF72* antisense RNAs and/or altered downstream molecular pathways hold great promise for these devastating neurodegenerative diseases.

## Materials and methods

### Plasmids and siRNAs

A construct containing 10 GGGGCC repeats, flanked 5′ by BbsI and 3′ by BsmBI recognition sites, was synthesized by GENEWIZ and used to generate antisense C$_4$G$_2$ repeats using recursive directional ligation as previously described (*Mizielinska et al., 2014*). The repeat-containing plasmids were amplified using recombination-deficient Stbl3 *Escherichia coli* (Life Technologies) at 32°C to minimize retraction of repeats. Human PKR cDNA was a gift from Dr. Thomas Dever (NIH, USA) and (C$_4$G$_2$)108RO was gifted by Dr. Adrian Isaacs (*Moens et al., 2018*). For longitudinal fluorescence microscopy, pGW1-mApple was used. All plasmids were verified by Sanger sequencing (Genewiz, USA). All MOE-Gapmer ASOs were synthesized by Integrated DNA Technologies, USA. The ASO sequences are sense ASO (GCCCCTAGCGCGCGACTC) (*Jiang et al., 2016*), AS ASO1 (GGCCGGGGCCGGGGCCGGGG), and

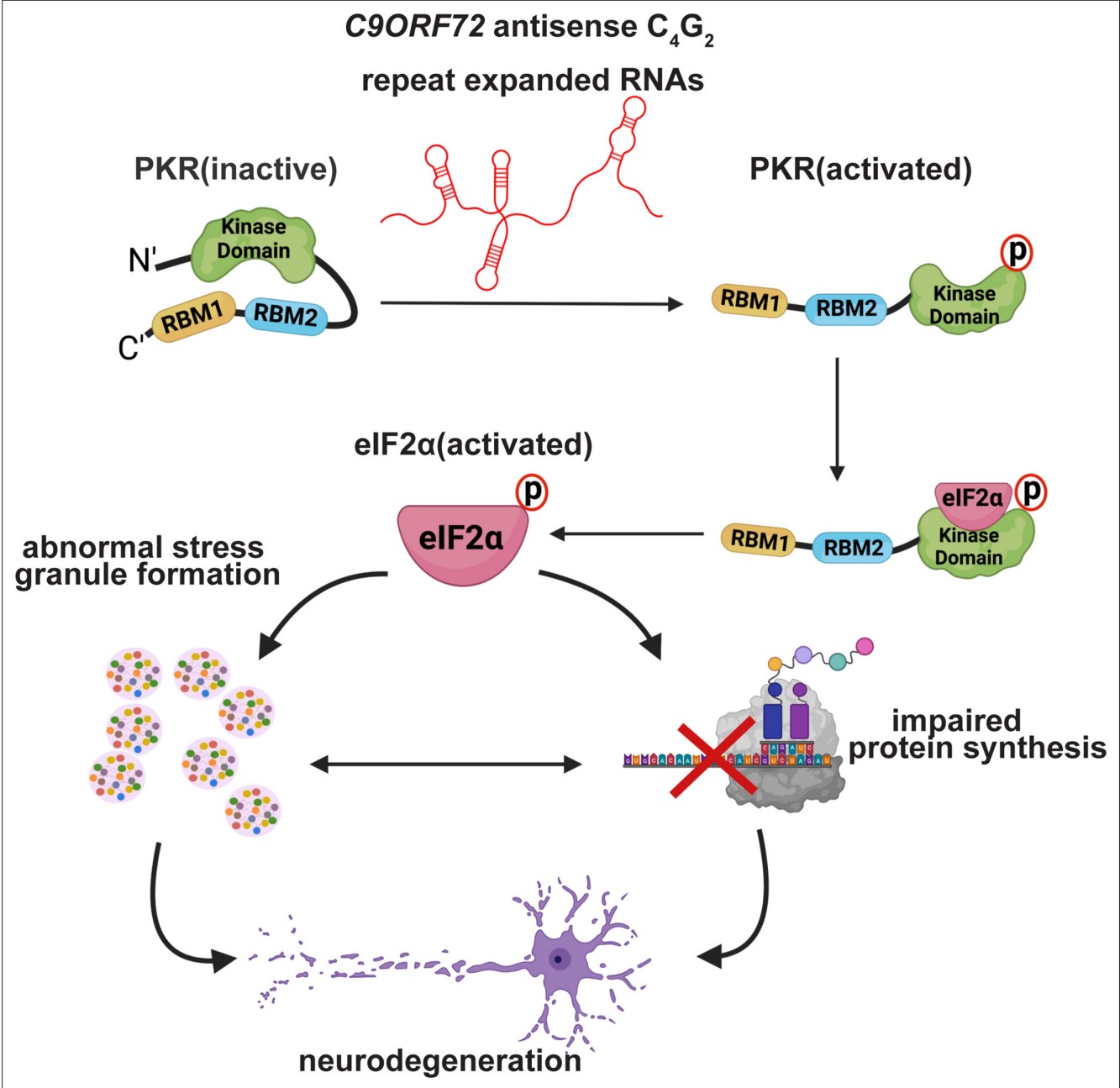

**Figure 8.** Proposed pathogenic mechanisms caused by *C9ORF72* antisense $C_4G_2$ repeat expanded RNAs. *C9ORF72* antisense $C_4G_2$ repeat expanded RNAs activate PKR/eIF2α-dependent integrated stress response and lead to neurotoxicity independent of DPR proteins.

AS ASO2 (CGGGGCCG GGGCCGGGGC). siRNAs against PKR and control siRNAs were purchased from Horizon Discovery, USA.

## Human tissues

Postmortem brain tissues from C9FTD/ALS patients (n = 6) and controls (n = 6) were obtained from the Emory Neuropathology Core. Patient information is provided in Table 1.

## Cell culture, transfection, and treatment

Human embryonic kidney (HEK293T) and human neuroblastoma (SH-SY5Y) cells from ATCC were cultured in high-glucose DMEM (Invitrogen) and DMEM-F12 (Invitrogen), respectively, supplemented with 10% fetal bovine serum (Corning), 4 mM GlutaMAX (Invitrogen), penicillin (100 U/mL), streptomycin

(100 µg/mL), and non-essential amino acids (1%). All parental cell lines were obtained from ATCC (HEK293T, CRL-3216; SH-SY5Y, CRL-2266) and are mycoplasma negative (abcam ab289834). Cells were grown at 37°C in a humidified atmosphere with 5% $CO_2$. Cells were transiently transfected using polyethyleneimine or Lipofectamine 2000. Experiments were performed 24 or 48 hr after transfection. For puromycin labeling, HEK293T and primary neurons were incubated with puromycin (10 ug/mL) for 30 and 5 min, respectively. For PKR inhibition, cells were treated with PKR inhibitor (C16) at 5 nM for 12 hr followed by transfection with repeats RNAs.

## Primary cortical neuronal culture and transfection

Primary cortical neurons were prepared from C57BL/6J mouse embryos (Charles River) of either sex on embryonic day 17. Cerebral cortices were dissected and enzymatically dissociated using trypsin with EDTA (Thermo Fisher Scientific; 10 min), mechanically dissociated in Minimum Essential Media (MEM; Fisher) supplemented with 0.6% glucose (Sigma) and 10% FBS (Hyclone) and stained to assess viability using Trypan Blue (Sigma). Neurons were plated on coverslips (Matsunami Inc, 22 mm). A total of 50,000 neurons were plated as a 'spot' on the center of the coverslip to create a small, high-density network. Neurons were cultured in standard growth medium (glial conditioned neurobasal plus medium [Fisher] supplemented with GlutaMAX [Gibco], and B27 plus [Invitrogen]), and half of the media was exchanged 2–3 times a week until the experiment endpoints. No antibiotics or anti-mycotics were used. Cultures were maintained in an incubator regulated at 37 °C, 5% $CO_2$, and 95% relative humidity as described (*Valdez-Sinon et al., 2020*). Cells were transiently transfected using Lipofectamine 2000 (Invitrogen) according to the manufacturer's instructions.

## Longitudinal fluorescence microscopy

Mouse primary cortical neurons were transfected with mApple and repeat expanded constructs and imaged by fluorescence microscopy at 24 hr intervals for 7–10 d as described (*Weskamp et al., 2019*). Time of death was determined based on rounding of the soma, retraction of neurites, or loss of fluorescence. The time of death for individual neurons was used to calculate the risk of death in each population relative to a reference group. Images were acquired using Keyence BZ-X810 microscope with a ×10 objective and analyzed using ImageJ. The images were stitched and stacked, and cell death was scored using the criteria mentioned above.

## RNA fluorescence in situ hybridization

LNA DNA probes were used against the sense and antisense hexanucleotide repeat expanded RNAs (Exiqon, Inc). The probe sequence for detecting sense RNA foci: TYE563-CCCCGGCCCCGGCCCC ; and that for antisense RNA foci is: TYE563-GGGGCCGGGGCCGGGG. All hybridization steps were performed under RNase-free conditions. Cells were fixed in 4% paraformaldehyde (Electron Microscopy Sciences) for 20 min, washed three times for 5 min with phosphate buffered saline (DEPC 1× PBS, Corning) followed by permeabilization with 0.2% Triton-X 100 (Sigma) for 10 min and then incubated with 2× SSC buffer for 10 min. Cells were hybridized (50% formamide, 2× SSC, 50 mM sodium phosphate [pH 7], 10% dextran sulfate, and 2 mM vanadyl sulfate ribonucleosides) with denatured probes (final concentration of 40 nM) at 66°C for 2 hr. After hybridization, slides were washed at room temperature in 0.1% Tween-20/2× SSC for 10 min twice and in stringency washes in 0.1× SSC at 65°C for 10 min. Cell nuclei were stained with DAPI. Three to six random pictures were taken by Keyence BZ-X810 microscope with a ×60 oil objective and analyzed using ImageJ.

## Immunofluorescence

Cells were fixed in 4% paraformaldehyde (Electron Microscopy Sciences) for 20 min, washed three times for 5 min with phosphate buffered saline (1×PBS, Corning), and treated with 0.2% Triton-X 100 (Sigma) in PBS for 10 min. Cells were blocked for 30 min in a blocking solution consisting of 4% bovine serum albumin (Sigma) in PBS. Cells were incubated overnight in primary antibodies diluted in blocking solution. The next day, cells were washed three times for 5 min in PBS and incubated in secondary antibodies in blocking solution for 1 hr at room temperature (dark). After washing three times for 5 min, coverslips with the cells were mounted using Prolong Gold Antifade mounting media (Invitrogen). Images were acquired with Keyence BZ-X810 microscope with a ×60 oil objective and analyzed using ImageJ.

## Immunohistochemistry

Postmortem brain tissues were obtained from the brain bank maintained by the Emory Alzheimer Disease Research Center under proper Institutional Review Board protocols. Paraffin-embedded sections from frontal cortex (8 μm thickness) were deparaffinized by incubation at 60°C for 30 min and rehydrated by immersion in graded ethanol solutions. Antigen retrieval was done by microwaving in a 10 mM citrate buffer (pH 6.0) for 5 min followed by allowing slides to cool to room temperature for 30 min. Endogenous peroxidase activity was eliminated by incubating slides with hydrogen peroxide block solution (Fisher) for 10 min at room temperature followed by rinsing in phosphate buffered saline. Nonspecific binding was reduced by blocking in ultra-Vision Block (Fisher) for 5 min at room temperature. Sections were then incubated overnight with primary antibodies diluted in 1% BSA in phosphate buffered saline for 30 min at room temperature or incubated without primary antibody as a negative control. Sections were rinsed in phosphate buffered saline and incubated in labeled ultra-Vision LP detection system horseradish peroxidase-polymer secondary antibody (Fisher) for 15 min at room temperature. Slides were imaged for analysis using an Aperio Digital Pathology Slide Scanner (Leica Biosystems). For IHC, rabbit anti-p-PKR, Millipore 07-532 (1:100 dilution) antibody was used.

## Protein lysate preparation

Whole cell/tissue extracts were lysed using RIPA Lysis Buffer pH 7.4 (Bio-world, USA) supplemented with Halt protease and phosphatase inhibitor cocktail (Thermo Fisher Scientific). Lysates were sonicated at 25% amplitude for three cycles for 15 s with 5 s intervals. Supernatant was collected after centrifuging at max speed for 15 min at 4°C. The concentration of the isolated proteins was determined using BCA Protein Assay Reagent (Pierce, USA).

## Immunoblotting assay

For western blotting, 20–30 μg of proteins were prepared in 4× Laemmli sample buffer and heat-denatured at 95°C for 5 min. Samples were resolved on 4–20% gradient gels (Bio-Rad). Proteins were transferred to nitrocellulose membranes (0.2 μm, Bio-Rad). The membrane was blocked in 5% milk and incubated overnight at 4°C with primary antibodies diluted in blocking buffer. Secondary antibodies HRP-conjugated secondary antibodies (ABclonal) or IRDye secondary antibodies (LI-COR) were diluted in blocking buffer and applied to the membrane for 1 hr at room temperature. Primary antibodies used mouse anti-FLAG (1:1000; Sigma # F1804), rabbit anti-HA (1:1000; Cell Signaling # 3724S), mouse anti-MYC (1:1000; Sigma # C3956), rabbit anti-PKR (1:1000; abcam # ab184257), rabbit anti-phospho-PKR (1:1000; abcam # ab32036), rabbit anti-eIF2α (1:1000; Cell Signaling # 3398S), rabbit anti-phospho-eIF2α (1:1000; CST# 9722S), rabbit anti-PERK (1:1000; CST # 3192S), rabbit anti-phospho-PERK (1:1000; abcam # ab192591), and rabbit anti-GAPDH (1:5000; ABclonal # ac001). Antibodies against PR, GP, and PA have been previously reported (*Jiang et al., 2016*). Super Signal West Pico (Pierce, USA) was used for detection of peroxidase activity. Molecular masses were determined by comparison to protein standards (Thermo Scientific). The immunoreactive bands were detected by ChemiDoc Image System (Bio-Rad, USA).

## Quantitative real-time PCR

Total RNAs were extracted using a RNeasy kit as instructed by the manufacturer (QIAGEN). cDNA was prepared using High-Capacity cDNA Reverse Transcription Kit from applied biosystem. Quantitative RT-PCR reactions were conducted and analyzed on a StepOnePlus Real-Time PCR system (Applied Biosystems). Gene expression levels were measured by SYBR green (Thermo Fisher Scientific) quantitative real-time PCR (PRIMER SEQUENCE).

## Zebrafish microinjections, SV2 immunohistochemistry, and phenotyping

Zebrafish work was performed as previously described (*Swinnen et al., 2018*). Zebrafish oocytes were injected at the one- to two-cell stage with the indicated amounts of morpholinos. The splice-blocking morpholino targeting exon 3 intron 3 junction of *Danio rerio eif2ak2* (transcript ENSDART00000164338.2; morpholino sequence 5′-AATGTCTTGAATACTGACC GGGTGA-3′), the translation-blocking morpholino targeting AUG start codon (morpholino sequence 5′-TTCCTGAC AGAGACTCCATTGCGAA-3′), and the standard control oligo (morpholino sequence 5′-CCTCTTAC CTCAGTTACAATTTATA-3′) were designed and generated by Gene Tools (Philomath, USA). Injected

oocytes were incubated at 28°C. After 24 hr post fertilization (hpf), the embryos were dechorionated using a forceps. Only morphologically normal embryos were selected for downstream experiments. At 30 hpf, the selected fish were deyolked and subsequently fixed overnight at 4°C in 4% formaldehyde in 1×PBS. Fish were permeabilized with acetone for 1 hr at − 20°C, blocked with 1% BSA/1% DMSO/PBS for 1 hr at room temperature, and immunostained with mouse anti-SV2 primary antibody (AB2315387, Developmental Studies Hybridoma Bank) followed by a secondary antibody.

For phenotyping, 15 embryos per condition were analyzed with imaging (Leica DM 3000 LED microscope; DMK 33UX250 USB3.0 monochrome industrial camera, The Imaging Source, Bremen, Germany) and the Lucia software (version 4.60, Laboratory Imaging, Prague, Czech Republic) by a blinded observer. For the axonal length, a standardized method was used; five predefined and consecutive motor axons (i.e. the 8th up to the 12th axon on one side) were measured in all 15 embryos. Data for axonal length were normalized to the control condition. For the abnormal branching, a predefined set of 20 consecutive motor axons (i.e. the 8th up to the 17th axon on both sides) in the same 15 embryos were analyzed. Motor axons were considered abnormal when axons branched at or before the ventral edge of the notochord. An embryo was considered as having 'abnormal branching' when at least two of these 20 axons were abnormal. For each experiment, the standard morpholino was used as control at the same dose of the tested morpholino.

## Statistical analysis

Statistical analyses and graphs were prepared in GraphPad Prism (version 9). Data is expressed as mean ± SD. Student's *t*-test or one-way ANOVA was used for statistical analysis unless specified in figure legends.

## Acknowledgements

We would like to thank Dr. Jonathan Glass for providing access to the clinical material (Emory cohort) and Dr. Homa Ghalei for thoughtful discussions. We would like to thank Ganesh Chilukuri and Daniel Pun for helping with the neuronal cell death counting. We are grateful to Dr. Yao Yao (University of Georgia) and Dr. Zachary McEachin (Emory University) for providing sense and control ASOs, respectively. JP is supported by the Milton Safenowitz Postdoctoral Fellowship from the ALS association (21-PDF-585). DCP is supported by Milton Safenowitz Postdoctoral Fellowship from the ALS association (22-PDF-605). AB and GJB are supported by the NIH R01 (R01NS114253 to GJB). EB is supported by a PhD Fellowship from FWO-Vlaanderen (1145621N). LVDB is supported by a research project of FWO-Vlaanderen (G0C1620N). The work was supported by the NIH R01 grant (R01AG068247 to JJ) and NIH R21 grant (5R21NS114908-02 to GJB and JJ).

## Additional information

### Funding

| Funder | Grant reference number | Author |
| --- | --- | --- |
| National Institute of Neurological Disorders and Stroke | R01NS114253 | Anwesha Banerjee Gary J Bassell |
| National Institutes of Health | R01AG068247 | Jie Jiang |
| ALS Association | 21-PDF-585 | Janani Parameswaran |
| Fonds Wetenschappelijk Onderzoek | G0C1620N | Ludo Van Den Bosch |
| ALS Association | 22-PDF-605 | Devesh C Pant |
| NIH | R01NS114253 | Gary J Bassell |
| FWO-Vlaanderen | 1145621N | Elke Braems |
| FWO-Vlaanderen | G0C1620N | Ludo Van Den Bosch |

| Funder | Grant reference number | Author |
| --- | --- | --- |
| NIH | 5R21NS114908-02 | Gary J Bassell<br>Jie Jiang |

The funders had no role in study design, data collection and interpretation, or the decision to submit the work for publication.

## Author contributions

Janani Parameswaran, Conceptualization, Resources, Formal analysis, Validation, Investigation, Methodology, Writing - original draft, Project administration, Writing - review and editing; Nancy Zhang, Data curation, Validation, Methodology; Elke Braems, Formal analysis, Methodology; Kedamawit Tilahun, Formal analysis, Validation; Devesh C Pant, Emma Davis, Investigation; Keena Yin, Seneshaw Asress, Kara Heeren, Methodology; Anwesha Banerjee, Samantha L Schwartz, Graeme L Conn, Gary J Bassell, Resources; Ludo Van Den Bosch, Resources, Supervision, Investigation; Jie Jiang, Resources, Supervision, Funding acquisition, Investigation, Methodology, Writing - original draft, Project administration, Writing - review and editing

## Author ORCIDs

Janani Parameswaran (iD) http://orcid.org/0000-0001-9030-4953
Kedamawit Tilahun (iD) http://orcid.org/0000-0003-2107-1580
Devesh C Pant (iD) http://orcid.org/0000-0003-4046-4195
Jie Jiang (iD) http://orcid.org/0000-0001-9519-4992

## Decision letter and Author response

Decision letter https://doi.org/10.7554/eLife.85902.sa1
Author response https://doi.org/10.7554/eLife.85902.sa2

# Additional files

## Supplementary files

• MDAR checklist

## Data availability

All data generated or analysed during this study are included in the manuscript.

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
