## [Editor Report]

The current study provides important, mechanistic insight into the potential contribution of antisense C4G2 expanded RNA to disease in C9orf72-associated ALS/FTD. The authors convincingly demonstrate that expression of this RNA species activates the PKR/eIF2α-dependent integrated stress response. They further provide evidence that this can contribute to disease phenotypes using multiple models and post-mortem patient samples.

---

## [Decision Letter]

**Decision letter after peer review:**

Thank you for submitting your article "Antisense, but not sense, repeat expanded RNAs activate PKR/1 eIF2α-dependent ISR in C9ORF72 FTD/ALS" for consideration by *eLife*. Your article has been reviewed by 2 peer reviewers, and the evaluation has been overseen by a Reviewing Editor and Suzanne Pfeffer as the Senior Editor. The following individuals involved in the review of your submission have agreed to reveal their identity: Roy Parker (Reviewer #1); Giovanna Mallucci (Reviewer #2).

Essential revisions:

1) Can the authors add data demonstrating that the sense transcript is or is not produced from their antisense C4G2 cassette as per reviewer comments?

2) Can the authors demonstrate that protein translation is rescued upon PKR knockdown as per reviewer comments?

3) The text would be improved by adding additional insight into the potential mechanism of PKR activation by the antisense strand. The hypothesis that the effect is direct and dependent on the secondary structures formed by the RNA is interesting but, as highlighted by reviewer comments, is not the only potential mechanism underlying the data and is not as clearly supported by the current literature as described. The effect could be indirect, dependent on the expression level of the RNAs, involve TDP-43, etc. Alternatively, the authors could provide data supporting their current hypothesis that the effect is dependent on the structure of the C4G2 RNA while this is likely more fitting for future investigations.

4) Please add more information on the ASOs utilized in the study.

*Reviewer #1 (Recommendations for the authors):*

1. It is not clear whether the antisense C4G2 cassette used at the beginning of the paper also produces a sense G4C2 transcript. While the authors do demonstrate in the case of the G4C2 sense cassette that the antisense transcript is what causes PKR activation, it would be nice for the readers to know whether the G4C2 sense transcript is made from the C4G2 antisense cassette.

2. I could not find information on the exact sequence of the ASO used. Please provide.

3. There is a disconnect between the text and the data in Figure 6 supplement 1F. I assume it is because of a mislabeling of the figure? Currently, the data suggest that treatment of (G4C2)108RO with BOTH antisense and sense ASOs leads to elevated p-PKR.

4. As the authors mention in the discussion, it has previously been shown that phosphorylated PKR is elevated in C9ORF72 ALS/FTD. However, the authors should also make explicit reference to this work in the Results section when showing the p-PKR brain histology.

5. In the discussion, the authors state that sense G4C2 RNAs form G-quadruplexes. The work from Wang et al., 2019, however, indicate that longer G4C2 likely form hairpin structures.

*Reviewer #2 (Recommendations for the authors):*

Beautifully written and well-presented overall.

It would be good to show the rescue of protein synthesis reduction by PKRi/knockdown in HEKs as done for all other readouts (Figure 2A-C)

Ultimately, it would be good to know (1) the mechanism by which anti-sense secondary structures activate PKR and (2) the relative contributions of this toxic mechanism and others, but this is for future work.

---

## [Author Response]

Reviewer #1 (Recommendations for the authors):1. It is not clear whether the antisense C4G2 cassette used at the beginning of the paper also produces a sense G4C2 transcript. While the authors do demonstrate in the case of the G4C2 sense cassette that the antisense transcript is what causes PKR activation, it would be nice for the readers to know whether the G4C2 sense transcript is made from the C4G2 antisense cassette.

We thank the reviewer for the pointing out the excellent question. We have included results in the revised manuscript (Line 272-273, Figure 6—figure supplement 1C). Interestingly, we also detected accumulation of sense foci in cells expressing antisense (C4G2)75 in HEK293T cells. We have discussed this new finding in line 407-415.

2. I could not find information on the exact sequence of the ASO used. Please provide.

Thank you for the comment. We have now included more information on ASOs in the material and methods section (Line 454-457).

3. There is a disconnect between the text and the data in Figure 6 supplement 1F. I assume it is because of a mislabeling of the figure? Currently, the data suggest that treatment of (G4C2)108RO with BOTH antisense and sense ASOs leads to elevated p-PKR.

We apologize for the mistake and have corrected the mislabeling in Figure 6 supplement 1F.

4. As the authors mention in the discussion, it has previously been shown that phosphorylated PKR is elevated in C9ORF72 ALS/FTD. However, the authors should also make explicit reference to this work in the Results section when showing the p-PKR brain histology.

We appreciate the suggestion. We have now cited the reference work in the result section (Line 252-254).

5. In the discussion, the authors state that sense G4C2 RNAs form G-quadruplexes. The work from Wang et al., 2019, however, indicate that longer G4C2 likely form hairpin structures.

We agree with the reviewer and have corrected this in the discussion (Line 385-386). Additional work is needed to understand how antisense, but not sense, repeat expanded RNAs activate the PKR-eIF2a pathway, as also pointed out by the reviewer #2

Reviewer #2 (Recommendations for the authors):Beautifully written and well-presented overall.

We thank the reviewer for the comments.

It would be good to show the rescue of protein synthesis reduction by PKRi/knockdown in HEKs as done for all other readouts (Figure 2A-C)

Thank you for the suggestion. We have now demonstrated a slight but significant increase of global protein translation in (C4G2)108RO expressing cells upon PKR knockdown (Figure 3—figure supplement 1I).

Ultimately, it would be good to know (1) the mechanism by which anti-sense secondary structures activate PKR and (2) the relative contributions of this toxic mechanism and others, but this is for future work.

We totally agree with the reviewer and appreciate for the suggestion. We are currently investigating and working on both potential direct and indirect mechanisms of PKR activation by *the C9ORF72* antisense repeats. We have also further discussed this in the revised manuscript (Line 378-399, 425-427).